# Making Models Unmergeable via Scaling-Sensitive Loss Landscape

**Minwoo Jang** [1]  **Hoyoung Kim** [2]  **Jabin Koo** [3]  **Jungseul Ok** [1 3]

## Abstract

The rise of model hubs has made it easier to access reusable model components, making model merging a practical tool for combining capabilities. Yet, this modularity also creates a *governance gap*: downstream users can recompose released weights into unauthorized mixtures that bypass safety alignment or licensing terms. Because existing defenses are largely post-hoc and architecture-specific, they provide inconsistent protection across diverse architectures and release formats in practice. To close this gap, we propose TRAP$^2$, an architecture-agnostic protection framework that encodes protection into updates during fine-tuning, regardless of whether they are released as adapters or full models. Instead of relying on architecture-dependent approaches, TRAP$^2$ uses weight re-scaling as a simple proxy for the merging process. It keeps released weights effective in standalone use, but degrades them under re-scaling that often arises in merging, undermining unauthorized recomposition.

## 1. Introduction

Public model hubs and open repositories, such as GitHub and Hugging Face, widely distribute fine-tuned updates, from full checkpoints to lightweight adapters. Since many updates target the same base model, they can be recombined after release by directly composing their parameters. This accessibility enables model merging, but it also creates a *governance gap*: once released, these updates can be recomposed into unauthorized mixtures that bypass safety, licensing, or task-specific constraints. In other words, facilitating broad reuse inherently reduces the ability to enforce post-release constraints.

[1]Graduate School of AI, POSTECH, Pohang, Republic of Korea [2]National AI Research Lab, Seoul, Republic of Korea [3]Department of CSE, POSTECH, Pohang, Republic of Korea. Correspondence to: Jungseul Ok <jungseul@postech.ac.kr>.

*Proceedings of the $43^{rd}$ International Conference on Machine Learning*, Seoul, South Korea. PMLR 306, 2026. Copyright 2026 by the author(s).

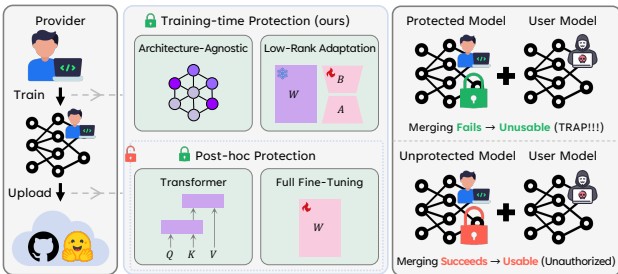

*Figure 1.* Unmergeability protection in model sharing and limitations of prior work. (Left) Providers release task updates for downstream reuse, often as adapters. (Middle) Most post-hoc protections are Transformer-specific, limiting transferability. They also require full-weight access, making them incompatible with adapter-only releases such as LoRA. (Right) These limitations motivate training-time protection embedded in the released update, preserving standalone utility while disrupting downstream merges.

This loss of control motivates the notion of *unmergeability* (Junhao et al., 2025; Wang et al., 2025). Ideally, a released model should retain full utility in its standalone setting, while failing reliably when incorporated into unauthorized merges. However, achieving this goal is challenging, because merging is performed downstream, outside the control of the creator, and often under heterogeneous protocols.

Most existing defenses address this challenge in a *post-hoc* manner: they apply function-preserving transformations to disrupt merging without changing the fine-tuning pipeline (Junhao et al., 2025; Wang et al., 2025). However, most existing methods are tailored to architectural symmetries of Transformers (Vaswani et al., 2017), which limits their transferability to non-Transformer backbones. Moreover, they assume access to the *full* model weights, making them mismatched with hub-style releases where only adapter updates, exemplified by Low-Rank Adaptation (LoRA) (Hu et al., 2022), are shared and the base weights are unavailable. Figure 1 summarizes this model-sharing workflow and highlights these two gaps in existing post-hoc defenses.

Against this background, we ask: *Can unmergeability be embedded directly into fine-tuned parameters across architectures and release formats?* We answer yes with TRAP$^2$ (**Tra**ining-time **P**rotection via **T**ask-**R**obust **A**dversarial **P**erturbation), which learns a protected update during fine-tuning. Conceptually, TRAP$^2$ optimizes an adversarial objective over update re-scaling: as illustrated in Figure 2,

it preserves utility at the authorized scale ($s = 1$), while inducing degradation under unauthorized scaling ($s \neq 1$), a regime that frequently appears in merging pipelines. Consequently, $\text{TRAP}^2$ yields brittleness under merging for both adapter-only and full-checkpoint releases, without relying on architecture-specific assumptions.

In summary, our contributions are as follows:

- **Problem Setup:** We formalize a post-release protection setting for fine-tuned releases across various architectures and release formats (adapter-only updates and full checkpoints), together with a unified evaluation protocol that quantifies standalone utility and degradation under merging.

- **Protection Method:** We introduce $\text{TRAP}^2$, a training-time procedure that keeps a fine-tuned update effective at the nominal scale ($s = 1$), while making it brittle under *off-nominal re-scaling* ($s \neq 1$), which captures the re-weighting effects commonly introduced by practical merging pipelines.

- **Empirical and Theoretical Analysis:** We evaluate $\text{TRAP}^2$ across diverse merging operators, release formats, and architectures, and complement the empirical results with theoretical analysis, establishing (i) convergence under stochastic optimization and (ii) degradation under down-scaling and model merging.

## 2. Related Works

### 2.1. Model Merging

Model merging aims to compose multiple task-specific models into a single model, with little or no additional training. Early research has focused on full-model merging, where multiple checkpoints are linearly combined. Representative methods include Task Arithmetic (TA) (Ilharco et al., 2023), TIES-Merging (Yadav et al., 2023), and DARE (Yu et al., 2024), which aggregate task updates via weighted summation, often coupled with pruning or sign-conflict resolution.

In parallel, Low-Rank Adaptation (LoRA) (Hu et al., 2022) has emerged as the de facto standard for parameter-efficient fine-tuning. As LoRA adapters are widely shared, merging has naturally extended to the adapter setting. While general merging operators are often directly applied to adapters, recent works, such as KnOTS (Stoica et al., 2025) and Core Space (Panariello et al., 2025), propose merging schemes specifically tailored for low-rank subspaces.

While these developments make downstream reuse easier, they also widen a *governance gap*. Post-release merging can obscure provenance and dilute license, IP, or safety constraints by blending protected adapters into composite models (Cong et al., 2024; Xu et al., 2025b; Rosati et al.,

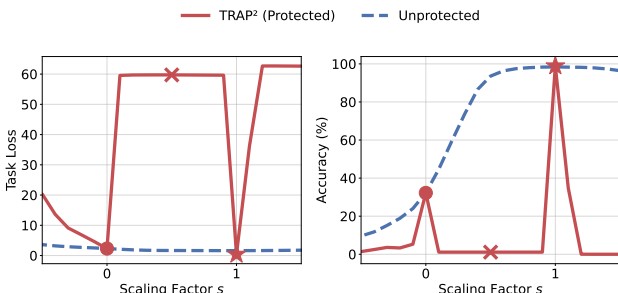

*Figure 2.* (Left) Loss shaping over the scaling factor $s$. We optimize via $\text{TRAP}^2$ to retain high utility in the *authorized* scale ($\star$; $s = 1$), while inducing degradation under *unauthorized* scales ($\times$; $s \neq 1$). The zero-shot result ($\bullet$; $s = 0$) is shown as a reference. (Right) Accuracy along the scaling factor $s$. $\text{TRAP}^2$-trained adapter attains high standalone accuracy in the authorized region ($\star$) but collapses under unauthorized scaling ($\times$).

2024; Hammoud et al., 2024). This vulnerability motivates protection mechanisms that preserve standalone utility while making unauthorized recomposition unreliable.

### 2.2. Unmergeability for Model Protection

Unmergeability aims to keep released weights useful in isolation, while making unauthorized merging unreliable. This capability is vital for model supply-chain control, since recomposition can obscure provenance and complicate the enforcement of usage terms. Existing defenses typically take a *post-hoc* approach, preserving standalone behavior via function-preserving parameter transformations.

PaRaMS (Junhao et al., 2025) and Merge-Lock (Wang et al., 2025) instantiate this idea for Transformers (Vaswani et al., 2017). They leverage architectural symmetries via coupled transformations of attention projections, preserving standalone behavior, while reducing compatibility under direct weight-space merging. To maintain functional equivalence, they must reparameterize the full set of weights, which makes them incompatible with adapter-only releases (e.g., LoRA) and less transferable to non-Transformer backbones.

In contrast, we target common release settings where full-weight access is unavailable or non-Transformer backbones are used. Accordingly, we propose a training-time mechanism that embeds unmergeability directly into the fine-tuned update, applicable across architectures and release formats, including adapter-only and full-checkpoint releases.

## 3. Background

We briefly review merging protocols to establish the scaling and aggregation notation used in Sections 4 and 5.

**Low-Rank Adaptation (LoRA)** Given a pre-trained model with parameters $W_0 \in \mathbb{R}^{d_{\text{out}} \times d_{\text{in}}}$, LoRA (Hu et al., 2022) freezes $W_0$ and injects trainable low-rank updates into

selected linear layers. Concretely, for $r \ll \min\{d_{\text{out}}, d_{\text{in}}\}$,

$$\Delta W = BA, \quad B \in \mathbb{R}^{d_{\text{out}} \times r}, \quad A \in \mathbb{R}^{r \times d_{\text{in}}}, \quad (1)$$

and the forward-pass weight becomes $W = W_0 + s \cdot \Delta W$, where $s \in \mathbb{R}_{\geq 0}$ denotes a scaling factor applied to the update $\Delta W$. The resulting parameters are lightweight and modular, and are commonly released as standalone adapters.

**LoRA Merging**   Given $N$ task-specific LoRA adapters $\{\Delta W_i\}_{i=1}^{N}$ trained on the same base model $W_0$, LoRA merging aims to compose them into a single adapter without additional training. The most common approach computes a linear aggregation of updates (Ilharco et al., 2023),

$$\Delta W_{\text{merged}} = \sum_{i=1}^{N} s_i \cdot \Delta W_i, \quad (2)$$

where $s_i$ denotes the merging coefficient for the $i^{\text{th}}$ adapter. While recent methods such as KnOTS (Stoica et al., 2025) and Core Space (Panariello et al., 2025) introduce structured subspace projections or alignment steps, they rely on aggregating updates that are (approximately) aligned. We provide a detailed overview of these techniques in Appendix B.

**Unified View: Merging as Aggregating Updates**   This perspective also covers full-checkpoint merging. Given checkpoints $\{W_i\}_{i=1}^{N}$ derived from the same base $W_0$, methods such as TA (Ilharco et al., 2023), TIES-Merging (Yadav et al., 2023), and DARE (Yu et al., 2024) first form task updates $\Delta W_i := W_i - W_0$ and then merge them (often with pruning or sign-conflict resolution) as

$$W_{\text{merged}} = W_0 + \sum_{i=1}^{N} s_i \cdot \Delta W_i = W_0 + \Delta W_{\text{merged}}. \quad (3)$$

Throughout this paper, we use $\Delta W$ to denote a generic released *update* (either a LoRA adapter or a full-parameter difference). We use $s$ for update scaling, and $s_i$ for coefficients used to aggregate updates.

## 4. Problem Setup and Challenges

We formalize *unmergeability* under hub-style model sharing and explain why existing post-hoc defenses apply to neither non-Transformer architectures nor adapter releases.

**Loss under Scaling**   Let $\mathcal{D}$ be a data distribution over examples $\xi$. We define the expected loss at weights $W$ as

$$\mathcal{L}(W; \mathcal{D}) := \mathbb{E}_{\xi \sim \mathcal{D}}[\ell(W; \xi)]. \quad (4)$$

When $\mathcal{D}$ is clear from context, we omit it and write $\mathcal{L}(W)$. For an update $\Delta W$ and a scaling factor $s \in \mathbb{R}_{\geq 0}$, define

$$\begin{aligned} \mathcal{L}_{\text{scaled}}(\Delta W; s) &:= \mathcal{L}(W_0 + s \cdot \Delta W) \\ &= \mathbb{E}_{\xi \sim \mathcal{D}}[\ell(W_0 + s \cdot \Delta W; \xi)]. \end{aligned} \quad (5)$$

We take $s = 1$ as the *nominal* (standalone) scale, and interpret $s \neq 1$ as off-nominal re-scaling induced by downstream reuse and merging. Accordingly, the intended composition is $W_0 + \Delta W$, and $\mathcal{L}_{\text{nominal}}(\Delta W) := \mathcal{L}_{\text{scaled}}(\Delta W; 1)$. In Section 5, we will also consider off-nominal scales ($s \neq 1$).

**Merging Operator**   Let $\mathcal{M}(\cdot)$ denote a merging operator that composes a set of updates into a single merged one. Given $N$ adapters $\{\Delta W_i\}_{i=1}^{N}$ trained on the same base $W_0$, the merged model is parameterized by

$$W_0 + \Delta W_{\text{merged}}, \qquad \Delta W_{\text{merged}} = \mathcal{M}\left(\{\Delta W_i\}_{i=1}^{N}\right). \quad (6)$$

A common instance is linear combination with scaling (Ilharco et al., 2023), as defined in Eq. (2).

**Desired Properties**   Our goal is to obtain an update $\Delta W^{\star}$ that satisfies the following properties established in prior works (Junhao et al., 2025; Wang et al., 2025).

*(Property 1) Standalone Utility:*   When deployed alone, $\Delta W^{\star}$ should incur low loss, i.e., $\mathcal{L}_{\text{nominal}}(\Delta W^{\star})$ is small.

*(Property 2) Unmergeability:*   Let $\mathcal{T} = \{\Delta W_i\}_{i=1}^{N-1}$ be third-party adapters and $\Delta W_{\text{merged}} := \mathcal{M}(\{\Delta W^{\star}\} \cup \mathcal{T})$. Define $\mathcal{L}_{\text{merged}}(\Delta W^{\star}; \mathcal{T}) := \mathcal{L}(W_0 + \Delta W_{\text{merged}})$. We require that indiscriminate merging yields a substantially worse merged model than merging an *unprotected* update of comparable standalone utility, i.e.,

$$\mathcal{L}_{\text{merged}}(\Delta W^{\star}; \mathcal{T}) \gg \mathcal{L}_{\text{merged}}(\Delta W'; \mathcal{T}), \quad (7)$$

where $\Delta W'$ is an unprotected adapter for the same task with $\mathcal{L}_{\text{nominal}}(\Delta W') \approx \mathcal{L}_{\text{nominal}}(\Delta W^{\star})$.

**Post-hoc Defenses via Paired Cancellations**   Most prior unmergeability defenses are *post-hoc*: they apply function-preserving re-parameterizations to the *full* model weights, preserving standalone behavior while disrupting direct weight-space merging. Two common symmetry templates are: **(i) FFN hidden-unit permutation**, which permutes intermediate neurons and compensates in adjacent linear layers; and **(ii) coupled self-attention reparameterization**, which exploits cancellation structures in Transformers (Vaswani et al., 2017). Concretely, with $Q = XW_Q$, $K = XW_K$, and $V = XW_V$, the attention output is

$$\text{Attention}(X) = \text{softmax}\left(\frac{QK^{\top}}{\sqrt{d}}\right) VW_O. \quad (8)$$

A paired invertible re-parameterization preserves the computation by keeping $QK^{\top}$ and $VW_O$ invariant:

$$\begin{aligned} W_Q \mapsto W_Q R_1, \quad & W_K \mapsto W_K R_1^{-\top}, \\ W_V \mapsto W_V R_2, \quad & W_O \mapsto R_2^{-1} W_O, \end{aligned} \quad (9)$$

for invertible $(R_1, R_2)$.

**Algorithm 1** Pseudo-code of TRAP$^2$ (Adapter Version)

1: **Input:** fixed base model $W_0$, dataset $\mathcal{D}$, time step $T$, step sizes $\{\eta_t\}_{t=0}^{T-1}$, trade-off weight $\lambda$, scale range $[s_{\min}, s_{\max}]$, exclusion width $\delta$, weighting function $w(\cdot)$
2: Initialize a trainable LoRA adapter $\Delta W$
3: **for** $t = 0, 1, \ldots, T-1$ **do**
4:     Sample mini-batch $\mathcal{B}_t \sim \mathcal{D}$      $(\mathcal{B}_t = \{\xi_j\}_{j=1}^m)$
5:     $\mathcal{L}_{\text{nominal}} \leftarrow \frac{1}{m} \sum_{\xi \in \mathcal{B}_t} \ell(W_0 + \Delta W; \xi)$
6:     Draw $s \sim \text{Unif}([s_{\min}, 1-\delta] \cup [1+\delta, s_{\max}])$
7:     $\mathcal{L}_{\text{off}} \leftarrow \frac{1}{m} \sum_{\xi \in \mathcal{B}_t} w(s) \cdot \ell(W_0 + s \cdot \Delta W; \xi)$
8:     $J_t \leftarrow \mathcal{L}_{\text{nominal}} - \lambda \cdot \mathcal{L}_{\text{off}}$
9:     $\Delta W \leftarrow \Delta W - \eta_t \cdot \nabla_{\Delta W} J_t$
10: **end for**
11: **Output:** protected adapter $\Delta W^\star$

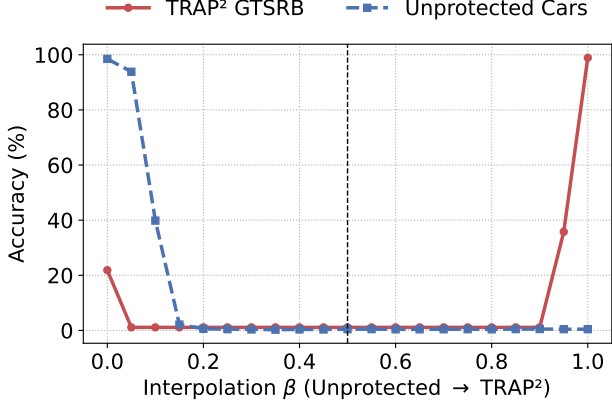

*Figure 3.* Performance degradation under pairwise merging. Accuracy along the interpolation path between an unprotected Cars adapter and a TRAP$^2$ GTSRB adapter is evaluated on both tasks.

PaRaMS (Junhao et al., 2025) combines MLP-level rearrangement with attention-level reweighting within this paired-cancellation framework. Merge-Lock (Wang et al., 2025) adopts a more expressive variant by composing random mixing, permutation, and diagonal reweighting to construct $R_1$ and $R_2$, while still relying on the same paired cancellations. In both cases, the function-preserving guarantee fundamentally assumes access to the full weight tensors on which the paired transformations act.

**Challenges** Modern hub ecosystems present two practical realities: (i) adapter-only release exemplified by LoRA, and (ii) architectural diversity beyond Transformers. These realities expose fundamental flaws in post-hoc defenses. First, applying paired transformations solely to the adapter $\Delta W$ fails to cancel terms associated with the frozen base model $W_0$. This misalignment either degrades standalone utility or fails to induce the intended unmergeability. Second, their symmetry templates are Transformer-specific and do not transfer to non-attention backbones such as ResNet (He et al., 2016) or ConvNeXt (Liu et al., 2022). These limitations motivate our central question: *Can we inject unmergeability directly into the released update, without relying on base-model access or architecture-specific symmetries?*

## 5. Proposed Method

To enforce unmergeability without sacrificing standalone utility, we propose TRAP$^2$, a training-time protection objective that directly shapes the released update. We first present the adapter (e.g., LoRA) instantiation in Algorithm 1. The central intuition is to regularize the adapter so that it remains accurate at the nominal scale ($s = 1$) yet degrades under off-nominal scaling ($s \neq 1$), a key effect commonly induced by adapter merging (e.g., the linear combination in Eq. 2).

**The TRAP$^2$ Objective** Let $\Delta W$ denote the trainable LoRA update on top of a fixed base $W_0$. For a deployment

scale $s \in \mathbb{R}_{\geq 0}$, let $\mathcal{L}_{\text{scaled}}(\Delta W; s)$ denote the expected loss at scale $s$, as defined in Eq. (5). The nominal loss corresponds to the intended scale $s = 1$:

$$\mathcal{L}_{\text{nominal}}(\Delta W) := \mathcal{L}_{\text{scaled}}(\Delta W; 1). \quad (10)$$

Let $\mathcal{S}$ be a distribution over off-nominal scales with support

$$\text{supp}(\mathcal{S}) \subseteq [s_{\min}, 1-\delta] \cup [1+\delta, s_{\max}], \quad (11)$$

where $\delta \in \mathbb{R}_{>0}$ specifies an exclusion margin around the nominal scale $s = 1$. To control how strongly we penalize different off-nominal scales, we introduce a nonnegative weighting function $w(s)$ and define the off-nominal loss as

$$\mathcal{L}_{\text{off}}(\Delta W) := \mathbb{E}_{s \sim \mathcal{S}}[w(s) \cdot \mathcal{L}_{\text{scaled}}(\Delta W; s)]. \quad (12)$$

Then, our goal is to minimize

$$J(\Delta W) = \mathcal{L}_{\text{nominal}}(\Delta W) - \lambda \cdot \mathcal{L}_{\text{off}}(\Delta W), \quad (13)$$

where $\lambda \in \mathbb{R}_{>0}$ trades off standalone utility and sensitivity to re-scalings. The first term preserves intended deployment performance, while the second term induces sensitivity to off-nominal re-scalings, encouraging unmergeability.

**Unmergeability Mechanism** Eq. (13) intentionally creates an asymmetry across various scales: it preserves performance at $s = 1$ by minimizing $\mathcal{L}_{\text{nominal}}(\Delta W)$, while increasing loss under off-nominal re-scalings via $\mathcal{L}_{\text{off}}(\Delta W)$. Because many merging operators effectively re-scale each constituent update (e.g., averaging scales each by $1/N$), the merged adapter is pushed away from its nominal operating point; TRAP$^2$ is trained to be brittle under such shifts. The weighting function $w(s)$ is used to normalize the training signal across scales. In particular, gradients can become small for down-scaled updates ($s < 1$), so we set $w(s) = 1/s$ by default to compensate for this effect and stabilize training.

*Table 1.* Standalone per-task accuracy (%; ↑) on 8 vision benchmarks for adapter-only deployment. We compare unprotected LoRA fine-tuning with post-hoc defenses (two LoRA variants of PaRaMS, Merge-Lock) and TRAP$^2$ across three CLIP backbones (ViT-B/32, ViT-L/14, ConvNeXt). For each dataset, bar length is normalized by the corresponding Fine-Tuned score (treated as 100), and bars below 95% of Fine-Tuned are colored pink. We omit the baselines for ConvNeXt since they are not directly applicable.

| Backbone | Method | Cars | DTD | EuroSAT | GTSRB | MNIST | RESISC | Aircraft | SVHN | Average |
|---|---|---|---|---|---|---|---|---|---|---|
| ViT-B/32 | Zero-Shot | 59.397 | 44.096 | 45.481 | 32.255 | 47.988 | 60.302 | 18.962 | 31.384 | 42.483 |
| | Fine-Tuned | 99.523 | 68.723 | 98.259 | 98.416 | 99.087 | 93.048 | 51.725 | 96.168 | 88.119 |
| | + Merge-Lock$^\star$ | 0.575 | 2.394 | 10.963 | 2.009 | 9.462 | 3.540 | 11.423 | 9.666 | 6.254 |
| | + Merge-Lock$^\dagger$ | 0.482 | 2.074 | 6.333 | 2.059 | 9.988 | 2.111 | 1.140 | 9.066 | 4.157 |
| | + PaRaMS$^\star$ | 97.839 | 67.713 | 98.296 | 98.238 | 99.125 | 92.476 | 51.305 | 96.111 | 87.638 |
| | + PaRaMS$^\dagger$ | 92.150 | 61.596 | 92.222 | 97.694 | 98.125 | 91.127 | 46.685 | 95.534 | 84.392 |
| | TRAP$^2$ (Ours) | 99.829 | 66.117 | 98.519 | 98.911 | 99.462 | 93.302 | 52.355 | 96.557 | 88.132 |
| ViT-L/14 | Zero-Shot | 77.864 | 55.532 | 62.259 | 50.713 | 76.162 | 71.365 | 32.583 | 58.552 | 60.629 |
| | Fine-Tuned | 99.767 | 76.702 | 98.778 | 98.466 | 99.587 | 95.587 | 72.577 | 97.758 | 92.403 |
| | + Merge-Lock$^\star$ | 0.497 | 2.340 | 8.593 | 3.256 | 11.737 | 1.571 | 0.930 | 6.473 | 4.425 |
| | + Merge-Lock$^\dagger$ | 0.389 | 1.862 | 9.111 | 1.880 | 11.950 | 1.635 | 1.110 | 8.259 | 4.525 |
| | + PaRaMS$^\star$ | 99.518 | 76.170 | 98.593 | 98.357 | 99.550 | 94.698 | 69.967 | 43.815 | 85.084 |
| | + PaRaMS$^\dagger$ | 98.912 | 74.043 | 98.407 | 98.248 | 99.638 | 93.397 | 67.027 | 41.424 | 83.887 |
| | TRAP$^2$ (Ours) | 99.876 | 78.830 | 98.481 | 99.584 | 99.438 | 96.175 | 80.138 | 97.167 | 93.711 |
| ConvNeXt | Zero-Shot | 89.554 | 59.628 | 54.519 | 48.892 | 54.888 | 67.143 | 27.512 | 34.327 | 54.558 |
| | Fine-Tuned | 98.492 | 76.436 | 98.926 | 99.268 | 99.325 | 96.063 | 59.226 | 97.105 | 90.605 |
| | TRAP$^2$ (Ours) | 97.606 | 76.064 | 98.630 | 98.931 | 99.488 | 94.079 | 64.326 | 96.159 | 90.660 |

**Optimization** As summarized in Algorithm 1, we optimize the TRAP$^2$ objective in Eq. (13) using stochastic first-order methods such as SGD. The expectation over $s \sim \mathcal{S}$ is approximated by a single-sample Monte Carlo draw per iteration, which provides an unbiased gradient estimator for $J$. In Appendix A.1, we provide a stationarity guarantee for this procedure under standard assumptions.

**Degradation from Down-Scaling** Uniformly averaging $N$ adapters re-scales each constituent adapter by a factor of $1/N$, placing the merged adapter away from the nominal scale. Since TRAP$^2$ explicitly amplifies loss under such off-nominal re-scalings, the merged one is expected to incur systematic degradation, as illustrated by the TRAP$^2$-trained GTSRB adapter in Figure 3. This effect is formalized in Appendix A.2, which characterizes the loss increase under down-scaling (e.g., $s = 1/N$ in the uniform averaging).

**Cross-Adapter Degradation** Merging a TRAP$^2$-trained adapter with an independently trained adapter moves the latter away from its nominal scale, as illustrated by the unprotected Cars adapter in Figure 3. Consequently, even an unprotected adapter can degrade after merging. Under mild regularity conditions, this degradation increases with the distance between the two adapters in parameter space. A formal analysis is deferred to Appendix A.4.

**Extension to Full Fine-Tuning** The same objective extends to full fine-tuning by defining a step-dependent update $\Delta W_t := W_t - W_0$. At each step $t$, we evaluate the nominal loss at $W_t$ and an off-nominal loss at the scaled model

$W_0 + s \cdot \Delta W_t$, and then update $W_t$. This yields a unified, architecture-agnostic formulation that applies to both adapter-based and full-checkpoint releases.

# 6. Experiments

In this section, we evaluate unmergeability under a realistic hub-style LoRA deployment setting, where a fixed base model is shared and only low-rank adapters are released and composed. Detailed settings (e.g., hyperparameters) for experiments can be found in Appendix D.

## 6.1. Baselines

We compare TRAP$^2$ against two post-hoc defenses, PaRaMS (Junhao et al., 2025) and Merge-Lock (Wang et al., 2025). Since they are not LoRA-native, we use two adapter-only variants: (i) an *adapter-space* variant ($\star$) that transforms the released update $\Delta W$ (for PaRaMS, this matches its LoRA setting using attention-head scaling only); and (ii) a *refitting* variant ($\dagger$) that transforms $W = W_0 + \Delta W$ and refits a rank-$r$ LoRA update via truncated SVD, which may incur approximation error. Detailed explanations for these baselines are provided in Appendix D.6.

## 6.2. Experiments on LoRA Merging

We consider eight standard image classification benchmarks: Cars (Krause et al., 2013), DTD (Cimpoi et al., 2014), EuroSAT (Helber et al., 2018), GTSRB (Stallkamp et al., 2012), MNIST (Deng, 2012), RESISC (Cheng et al., 2017), Aircraft (Maji et al., 2013), and SVHN (Netzer et al., 2011).

*Table 2.* Averaged per-task accuracy (%; ↓) under 8-way LoRA merging: in each trial, the target-task adapter is protected and merged with seven unprotected adapters using TA/TIES/TIES+DARE/TSV/CART in Full/KnOTS/Core spaces. Bars are normalized within each column, and values below 95% of *Unprotected* are highlighted in pink. We tune the merging coefficient via validation-based search over $s \in \{0.1, 0.2, \ldots, 10.0\}$ (including $s=1.0$), making results optimistic for the merger.

| Protection | Merging Method / Merging Space | | | | | | | | | | | | |
| --- | --- | --- | --- | --- | --- | --- | --- | --- | --- | --- | --- | --- | --- |
| | TA | TIES | | | TIES+DARE | | | TSV | | | CART | | |
| | Full | Full | KnOTS | Core | Full | KnOTS | Core | Full | KnOTS | Core | Full | KnOTS | Core |
| **Backbone:** **ViT-B/32** (Zero-shot: 42.483) | | | | | | | | | | | | | |
| *Unprotected* | 48.273 | 48.020 | 49.925 | 53.264 | 48.180 | 49.926 | 54.745 | 51.442 | 49.202 | 55.014 | 49.553 | 49.850 | 51.201 |
| PaRaMS* | 48.290 | 48.027 | 49.847 | 53.054 | 48.329 | 49.801 | 54.506 | 51.901 | 49.249 | 55.049 | 49.663 | 49.908 | 51.922 |
| PaRaMS† | 48.104 | 48.078 | 49.872 | 53.022 | 48.208 | 49.872 | 54.080 | 51.418 | 49.033 | 54.509 | 49.418 | 49.791 | 50.647 |
| TRAP² (Ours) | 23.121 | 36.480 | 37.514 | 37.213 | 33.880 | 28.383 | 36.752 | 24.815 | 24.210 | 24.449 | 41.324 | 40.967 | 41.602 |
| **Backbone:** **ViT-L/14** (Zero-shot: 60.629) | | | | | | | | | | | | | |
| *Unprotected* | 62.914 | 67.909 | 68.879 | 68.825 | 67.970 | 69.870 | 68.779 | 72.411 | 66.725 | 74.565 | 64.399 | 65.014 | 66.185 |
| PaRaMS* | 62.926 | 67.821 | 69.583 | 69.229 | 67.915 | 69.631 | 69.215 | 71.985 | 66.740 | 74.592 | 64.358 | 65.012 | 66.231 |
| PaRaMS† | 62.826 | 67.875 | 69.581 | 69.287 | 67.877 | 69.596 | 69.302 | 72.150 | 66.355 | 74.097 | 64.262 | 64.811 | 65.825 |
| TRAP² (Ours) | 33.619 | 53.162 | 47.749 | 54.462 | 50.630 | 43.269 | 54.847 | 41.993 | 37.369 | 39.389 | 58.634 | 58.444 | 59.514 |
| **Backbone:** **ConvNeXt** (Zero-shot: 54.558) | | | | | | | | | | | | | |
| *Unprotected* | 49.203 | 59.603 | 60.201 | 63.601 | 59.620 | 60.305 | 63.573 | 65.133 | 60.351 | 65.836 | 60.069 | 59.997 | 61.215 |
| TRAP² (Ours) | 14.719 | 32.883 | 15.187 | 17.188 | 15.636 | 13.054 | 25.335 | 14.085 | 15.850 | 15.296 | 46.973 | 45.885 | 47.204 |

We use CLIP (Radford et al., 2021) ViT-B/32 by default, and evaluate ViT-L/14 and ConvNeXt (Liu et al., 2022)-based CLIP models additionally. We fix the LoRA rank to $r = 16$ and fine-tune only LoRA parameters on a frozen base model. For each task, we train (i) an unprotected adapter, to which post-hoc baselines are applied after fine-tuning, and (ii) a TRAP² adapter trained under Eq. (13).

After training, adapters are merged using TA (Ilharco et al., 2023), TIES (Yadav et al., 2023), TIES+DARE (Yu et al., 2024), TSV (Gargiulo et al., 2025), and CART (Choi et al., 2024), in Full space as well as KnOTS (Stoica et al., 2025) and Core Space (Panariello et al., 2025). Unless otherwise specified, we tune the merging coefficient via a simple grid search starting from 0.1 with step size 0.1 and select the best value on the validation set, using early stopping with patience 10. This procedure always evaluates the nominal coefficient 1.0 and spans both down-scaling and up-scaling of constituent updates, so the reported results reflect an optimistic (strong) merger that searches for the most favorable coefficient under each operator and space. Details of the merging operators and spaces are provided in Appendix B.

**Standalone Performance** A practical protection mechanism must preserve standalone utility under adapter-only deployment. Table 1 reports per-task standalone accuracy on the eight vision benchmarks for TRAP² and the two post-hoc baselines, PaRaMS and Merge-Lock, using the two variants described in Section 6.1.

Merge-Lock collapses under both variants, often yielding accuracy close to chance level even before merging, making it unsuitable for adapter-only deployment. In contrast, PaRaMS and TRAP² retain standalone performance comparable to unprotected fine-tuning across tasks, satisfying the standalone-utility requirement.

**Performance Degradation under Merging** Standalone utility alone does not guarantee unmergeability. Indiscriminate merging should yield a degraded merged model, potentially impairing both the protected task and other merged tasks. Table 2 reports controlled merging results averaged over eight trials, where we protect one target adapter, train the remaining seven unprotected, and then merge all eight adapters under each operator and space.

We quantify unmergeability by comparing each protected merging to the corresponding unprotected merging under the same operator and space. PaRaMS exhibits only small deviations from *Unprotected* across datasets and operators, indicating that task performance largely survives standard merging. In contrast, TRAP² incurs substantial post-merge degradation, often pushing the accuracy below the zero-shot baseline. Complete results are provided in Appendix C.

**Scaling and Architecture Generalization** We further evaluate robustness to backbone scale and architecture by repeating the same protocol on CLIP ViT-L/14 and ConvNeXt-based CLIP models. On ViT-L/14, TRAP² preserves strong standalone performance while still inducing sharp post-merge degradation, whereas post-hoc baselines become unreliable under adapter-only deployment. On ConvNeXt, attention-symmetry-based defenses are not applicable, but TRAP² transfers without architectural assumptions and exhibits the same qualitative behavior, suggesting that the unmergeability effect persists across scale and architecture.

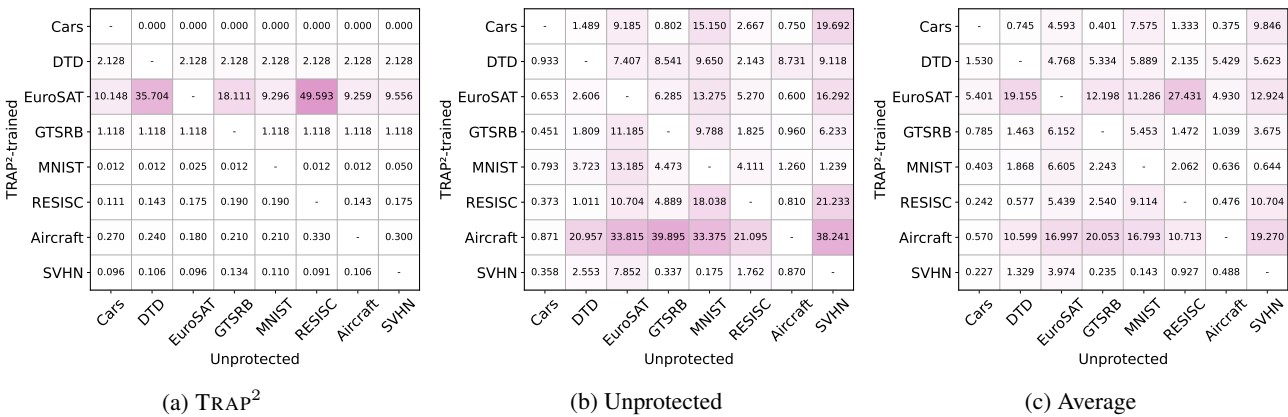

*Figure 4.* Results of pairwise LoRA merging at scale $s = 0.8$ on CLIP ViT-B/32. Each cell merges one TRAP$^2$-trained adapter for the row task with one unprotected adapter for the column task, and reports per-task accuracy (%; ↓) on (a) the protected task, (b) the unprotected task, and (c) their average. Cell color intensity encodes accuracy, with darker shading indicating lower accuracy (stronger degradation). The protected adapter consistently self-collapses after merging, often inducing collateral degradation on the unprotected task.

*Table 3.* Standalone per-task accuracy (%; ↑) on 8 vision benchmarks for CLIP ViT-B/32 under full fine-tuning. TRAP$^2$ preserves standalone performance comparable to standard fine-tuning.

| Backbone | Method | Cars | DTD | EuroSAT | GTSRB | MNIST | RESISC | Aircraft | SVHN | Average |
|---|---|---|---|---|---|---|---|---|---|---|
| | Zero-Shot | 59.397 | 44.096 | 45.481 | 32.255 | 47.988 | 60.302 | 18.962 | 31.384 | 42.483 |
| ViT-B/32 | Fine-Tuned | 99.767 | 65.000 | 98.630 | 97.615 | 99.212 | 93.175 | 51.785 | 95.602 | 87.598 |
| | TRAP$^2$ (Ours) | 99.798 | 62.766 | 98.444 | 99.109 | 99.212 | 93.317 | 49.625 | 96.706 | 87.372 |

*Table 4.* Averaged per-task accuracy (%; ↓) on 8 vision benchmarks for CLIP ViT-B/32 under full-model merging. TRAP$^2$ remains effective under full fine-tuning, inducing strong degradation after merging compared to the unprotected baseline.

| Protection | Merging Method | | | | |
|---|---|---|---|---|---|
| | TA | TIES | TIES+DARE | TSV | CART |
| **Backbone:** | **ViT-B/32** | (Zero-shot: 42.483) | | | |
| *Unprotected* | 49.963 | 50.951 | 51.673 | 62.419 | 61.031 |
| TRAP$^2$ (Ours) | 28.475 | 37.970 | 37.899 | 38.181 | 42.116 |

**Pairwise Merging**   Following prior works (Junhao et al., 2025; Wang et al., 2025), we merge one TRAP$^2$-trained adapter with one unprotected adapter and report per-task and average accuracy, as shown in Figure 4. Across task pairs, TRAP$^2$ consistently self-collapses: protected-task accuracy drops sharply, driving the decrease in average performance. We also observe collateral damage on unprotected tasks, consistent with Theorems A.2 and A.4. Taken together, TRAP$^2$ induces degradation even at the two-adapter level.

### 6.3. Experiments on Generalization

In this subsection, we ask whether the protection generalizes across three settings: full fine-tuning, mathematical reasoning tasks, and other LoRA variants.

**Full Fine-Tuning**   As discussed in Section 5, TRAP$^2$ extends beyond LoRA to full fine-tuning, by applying the objective on a step-dependent update $\Delta W_t := W_t - W_0$. We apply it on CLIP ViT-B/32 and merge the resulting checkpoints via standard weight averaging. Table 3 reports standalone performance under full fine-tuning, showing that TRAP$^2$ preserves utility comparable to the unprotected baseline. Table 4 then evaluates full-model merging, where the merged model degrades reliably, mirroring the LoRA setting and indicating that the effect is not LoRA-specific.

**Mathematical Reasoning Tasks**   Beyond vision classification, we apply TRAP$^2$ to Llama-3.1-8B (Grattafiori et al., 2024) with LoRA adapters of rank $r = 32$ on two mathematical reasoning benchmarks, GSM8K (Cobbe et al., 2021) and ASDiv (Miao et al., 2020), measured by Exact-Match (EM) accuracy. As shown in Figure 9 in Appendix C, standalone performance remains comparable to unprotected fine-tuning on both tasks, while pairwise interpolation involving a TRAP$^2$-protected adapter collapses sharply across a broad coefficient range, mirroring the vision setting.

**Other LoRA Variants**   To check whether the protection extends across LoRA variants, we evaluate TRAP$^2$ with QLoRA (Dettmers et al., 2023) and DoRA (Liu et al., 2024) on the GTSRB–Cars pair using the ViT-B/32 backbone. As shown in Figure 10 in Appendix C, standalone TRAP$^2$ matches unprotected fine-tuning under both variants, while pairwise interpolation with an unprotected adapter still collapses sharply across a broad coefficient range, indicating that the protection is not specific to standard LoRA.

*Table 5.* Averaged accuracy (%; ↓) under stronger threat models on CLIP ViT-B/32. TRAP² substantially degrades performance under both data-dependent merging (RegMean, CoM) and data-driven recovery (ProDistill, SFT), relative to the unprotected baseline.

| Protection | Data-dependent Merging | | Data-driven Recovery | |
|---|---|---|---|---|
| | RegMean | CoM | ProDistill | SFT |
| **Backbone:** | **ViT-B/32** | (Zero-shot: 42.483) | | |
| *Unprotected* | 49.107 | 64.776 | 73.823 | 56.859 |
| TRAP² (Ours) | 9.480 | 32.331 | 49.153 | 45.496 |

## 6.4. Robustness against Data-aware Methods

In this subsection, we conduct stress tests on TRAP² under stronger threat models with access to task-relevant data.

**Data-dependent Merging**  We evaluate TRAP² with two representative data-dependent mergers, RegMean (Jin et al., 2023) and Chain-of-Merges (CoM) (Buzzega et al., 2025), which use additional task-specific samples to compute aggregation weights for merging. As shown in Table 5, TRAP² induces substantial post-merge degradation under both Reg-Mean and CoM, suggesting that protection remains effective even when the merger leverages additional data signals. We also report the coefficients selected by CoM in Figure 5 as a diagnostic: unlike RegMean, CoM selects task-wise weights, and the coefficients show that it often assigns comparable or even larger weight to the TRAP²-protected adapter than to its unprotected counterpart, rather than down-weighting it to recover utility.

**Data-driven Recovery**  As a stronger threat model, we consider data-driven recovery, a white-box attack where the adversary uses a small task-relevant dataset to recover utility from a merged model. We evaluate two such procedures: ProDistill (Xu et al., 2025a), which learns merging coefficients via teacher-student distillation, and post-merge Supervised Fine-Tuning (SFT) of the merged model. As shown in Table 5, recovery from a TRAP²-protected merge gains only a few points above the zero-shot reference, whereas the unprotected baseline gains far more. These results suggest that TRAP² substantially impedes recovery even when the adversary has access to in-domain samples.

## 6.5. Additional Analysis

We provide additional analysis to test TRAP² with a uniform-averaging proxy that scales a single adapter by $s = 1/N$, and to quantify the per-step and end-to-end training cost, relative to standard LoRA fine-tuning.

**Uniform Averaging Proxy**  We test TRAP² with a uniform-averaging proxy, using update re-scaling as a lightweight pre-release check for merging behavior. For each task-specific adapter, we evaluate the scaling proxy

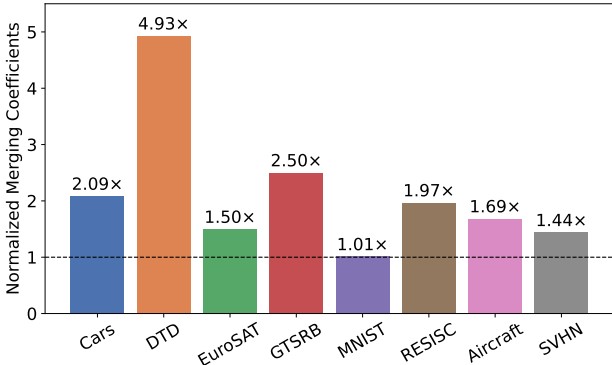

*Figure 5.* Normalized merging coefficients selected by CoM, when protecting each dataset-wise adapter with TRAP². Each coefficient is normalized by the corresponding unprotected baseline, showing that data-driven re-weighting of CoM does not prevent the collapse of TRAP²-protected adapters under merging.

$s = 1/N$, where $1/N$ corresponds to the coefficient assigned to each constituent update in a naive uniform average of $N$ adapters. Figure 6 reports the resulting accuracy across eight vision datasets in ViT-B/32, where each subplot directly compares TRAP² against the unprotected baseline within the same dataset. Across all datasets, TRAP² exhibits a consistent catastrophic degradation once $N \geq 2$, while the unprotected baseline remains largely stable as $N$ increases. These results indicate that TRAP² remains effective against indiscriminate uniform averaging, a common plug-and-play reuse pattern in open adapter ecosystems.

**Training Cost**  We quantify the training overhead introduced by the TRAP² objective, relative to standard LoRA fine-tuning. On a single RTX A6000 GPU, the additional sampling and gradient computation in Eq. (13) increase per-step time by $1.72\times$, while leaving peak GPU memory nearly unchanged. However, end-to-end wall-clock cost depends not only on per-step time but also on the number of optimization steps required to reach a target. As shown in Table 6, under a shared target accuracy TRAP² reaches the target faster on 5 out of 8 datasets, and under a shared wall-clock budget it achieves higher validation accuracy on 6 out of 8 datasets. These results indicate that the per-step overhead does not translate to a proportional end-to-end cost in practice. Indeed, TRAP² tends to require fewer optimization steps to reach a given target accuracy, partially offsetting the per-step overhead.

## 7. Conclusion

This paper studies protection of released fine-tuning updates against unauthorized post-release merging in model-sharing ecosystems. We propose TRAP², a training-time objective that embeds unmergeability directly into the released update by promoting sensitivity to off-nominal scaling, a lightweight proxy for the off-nominal coefficients

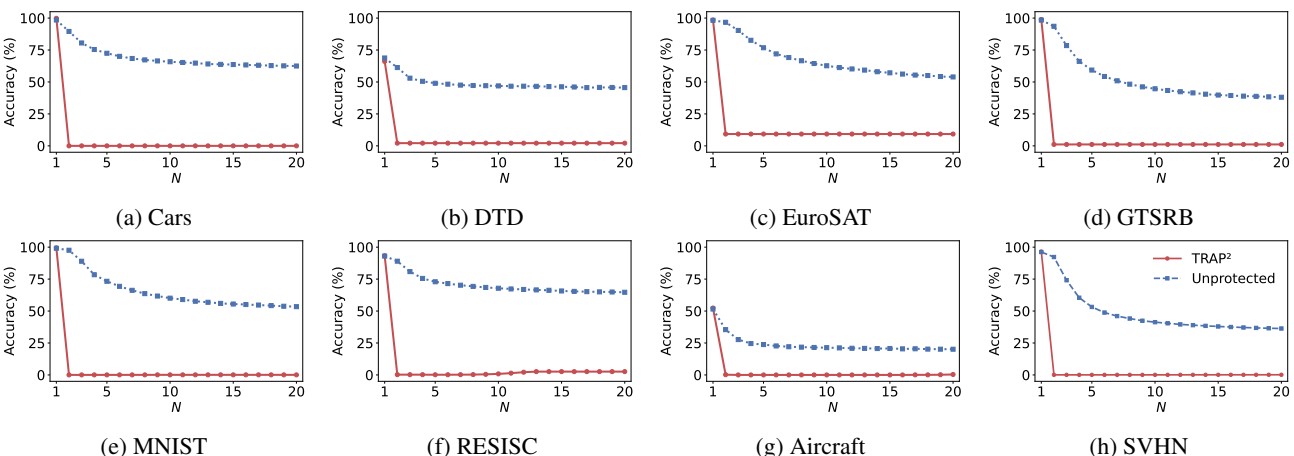

*Figure 6.* Uniform averaging proxy across 8 vision benchmarks. For each dataset, we evaluate a single task-specific adapter under the scaling proxy $s = 1/N$, where $1/N$ corresponds to the coefficient assigned to each constituent update in a naive uniform average of $N$ adapters. Thus, $N$ denotes a hypothetical averaging factor rather than an explicitly constructed $N$-way merge. We report accuracy (%) as a function of $N$ for each dataset. $\text{TRAP}^2$ exhibits catastrophic degradation as soon as $N \geq 2$, while unprotected adapters remain relatively stable, indicating robustness of the protection against uniform averaging.

*Table 6.* Training cost of $\text{TRAP}^2$ relative to unprotected LoRA on 8 ViT-B/32 adapters. **Speedup** ($\times$): wall-clock ratio (Unprotected / $\text{TRAP}^2$) to a shared target accuracy, where $> 1$ means $\text{TRAP}^2$ is faster. $\Delta$ **Acc** (pp): validation accuracy gap ($\text{TRAP}^2 -$ Unprotected) at a shared wall-clock budget, where $> 0$ means $\text{TRAP}^2$ is higher. $\text{TRAP}^2$ is faster on 5 out of 8 datasets and higher on 6 out of 8 datasets.

| Metric | Cars | DTD | EuroSAT | GTSRB | MNIST | RESISC | Aircraft | SVHN |
|---|---|---|---|---|---|---|---|---|
| Speedup ($\times$) | 0.268 | **2.097** | 0.661 | **4.264** | 0.763 | **2.694** | **6.097** | **2.269** |
| $\Delta$Acc (pp) | $-2.052$ | **+4.787** | $-0.296$ | **+2.811** | **+0.050** | **+0.904** | **+13.741** | **+0.768** |

and update transformations applied by common composition pipelines. Across experiments spanning architectures and release formats, $\text{TRAP}^2$ preserves strong standalone utility while consistently degrading performance under widely used merging operators, without requiring architecture-specific symmetries or access to full weights.

**Limitations** First, our claim of architecture-agnostic protection via $\text{TRAP}^2$ is objective- and formulation-level, rather than covering all architectures or deployment settings. Broader validation on larger and more diverse LLMs and multi-modal models remains an open direction. Second, the trade-off coefficient $\lambda$ requires per-dataset tuning. While this is a one-time pre-release cost rather than a deployment burden, automatic selection remains future work. Finally, our formulation assumes a shared base $W_0$ and targets a restrictive protection setting. This leaves base-model drift and permission-aware composition outside the current scope.

## Impact Statement

This paper proposes a training-time protection mechanism to discourage unauthorized or indiscriminate merging and to improve the integrity of adapter reuse in open ecosystems. Although our method uses adversarial perturbations during training, these are intended as a defensive tool for uploader protection, rather than to enable attacks. We do not anticipate negative societal impacts beyond those commonly associated with research on model security and responsible model sharing. However, as any protection mechanism may be misused (e.g., to disrupt third-party merging pipelines), we encourage responsible release practices such as clear labeling and usage guidelines. For example, a protected release can be accompanied by a short README stating that it is intended for standalone use and may not behave reliably under downstream merging.

## Acknowledgements

This work was partly supported by the IITP grants and the NRF grants funded by Ministry of Science and ICT, Korea (No.RS-2019-II191906, Artificial Intelligence Graduate School Program (POSTECH); No.RS-2024-00457882, AI Research Hub Project; RS-2024-00509258, Global AI Frontier Lab).

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

# A. Theoretical Analysis of $\text{TRAP}^2$

**Theorem A.1** (Stationarity of $\text{TRAP}^2$ under SGD). *Let $J(\Delta W)$ be the $\text{TRAP}^2$ objective defined in Eq. (13), with a nonnegative weight $w(s)$. Assume $J$ is $L$-smooth and bounded below by $J_{\inf} > -\infty$. Assume the scale distribution $\mathcal{S}$ is supported on two off-nominal intervals: for $0 < s_{min} < 1 - \delta < 1 + \delta < s_{max}$,*

$$\text{supp}(\mathcal{S}) \subseteq [s_{min}, 1 - \delta] \cup [1 + \delta, s_{max}].$$

*Consider SGD with step size $\eta \in (0, 1/L)$:*

$$\Delta W_{t+1} = \Delta W_t - \eta \, G_t.$$

*Let $\{(z_t, s_t)\}_{t \geq 0}$ be i.i.d. and define the filtration $\mathcal{F}_t := \sigma(\Delta W_0, (z_0, s_0), \ldots, (z_{t-1}, s_{t-1}))$. Assume the stochastic gradient satisfies, almost surely,*

$$\mathbb{E}[G_t \mid \mathcal{F}_t] = \nabla J(\Delta W_t), \qquad \mathbb{E}\big[\|G_t - \nabla J(\Delta W_t)\|_F^2 \mid \mathcal{F}_t\big] \leq \sigma_w^2.$$

*Then for any $T \geq 1$,*

$$\min_{0 \leq t \leq T-1} \mathbb{E}\|\nabla J(\Delta W_t)\|_F^2 \leq \frac{2(\mathbb{E}[J(\Delta W_0)] - J_{\inf})}{\eta T} + L\eta\sigma_w^2.$$

*In particular, with $\eta = \Theta(1/\sqrt{T})$, we obtain*

$$\min_{0 \leq t \leq T-1} \mathbb{E}\|\nabla J(\Delta W_t)\|_F^2 = O(1/\sqrt{T}).$$

*Here, $\sigma_w^2$ may depend on $w$ and $\text{supp}(\mathcal{S})$ (e.g., for $w(s) = 1/s$, it depends on $s_{min} > 0$).*

*Proof.* By $L$-smoothness of $J$, for any $t$,

$$J(\Delta W_{t+1}) \leq J(\Delta W_t) + \langle \nabla J(\Delta W_t), \Delta W_{t+1} - \Delta W_t \rangle + \frac{L}{2}\|\Delta W_{t+1} - \Delta W_t\|_F^2.$$

Using $\Delta W_{t+1} - \Delta W_t = -\eta G_t$ yields

$$J(\Delta W_{t+1}) \leq J(\Delta W_t) - \eta \langle \nabla J(\Delta W_t), G_t \rangle + \frac{L\eta^2}{2}\|G_t\|_F^2.$$

Taking conditional expectation given $\mathcal{F}_t$ and using $\mathbb{E}[G_t \mid \mathcal{F}_t] = \nabla J(\Delta W_t)$, we have

$$\mathbb{E}[\langle \nabla J(\Delta W_t), G_t \rangle \mid \mathcal{F}_t] = \langle \nabla J(\Delta W_t), \mathbb{E}[G_t \mid \mathcal{F}_t] \rangle = \|\nabla J(\Delta W_t)\|_F^2.$$

Moreover, by the conditional bias–variance decomposition,

$$\begin{aligned}
\mathbb{E}[\|G_t\|_F^2 \mid \mathcal{F}_t] &= \|\mathbb{E}[G_t \mid \mathcal{F}_t]\|_F^2 + \mathbb{E}\big[\|G_t - \mathbb{E}[G_t \mid \mathcal{F}_t]\|_F^2 \mid \mathcal{F}_t\big] \\
&= \|\nabla J(\Delta W_t)\|_F^2 + \mathbb{E}\big[\|G_t - \nabla J(\Delta W_t)\|_F^2 \mid \mathcal{F}_t\big] \\
&\leq \|\nabla J(\Delta W_t)\|_F^2 + \sigma_w^2.
\end{aligned}$$

Substituting these bounds gives

$$\mathbb{E}[J(\Delta W_{t+1}) \mid \mathcal{F}_t] \leq J(\Delta W_t) - \eta\left(1 - \frac{L\eta}{2}\right)\|\nabla J(\Delta W_t)\|_F^2 + \frac{L\eta^2}{2}\sigma_w^2.$$

Since $\eta \leq 1/L$, we have $1 - \frac{L\eta}{2} \geq \frac{1}{2}$, so that

$$\mathbb{E}[J(\Delta W_{t+1}) \mid \mathcal{F}_t] \leq J(\Delta W_t) - \frac{\eta}{2}\|\nabla J(\Delta W_t)\|_F^2 + \frac{L\eta^2}{2}\sigma_w^2.$$

Taking total expectation and summing over $t = 0, \ldots, T - 1$ yields

$$\frac{\eta}{2}\sum_{t=0}^{T-1} \mathbb{E}\|\nabla J(\Delta W_t)\|_F^2 \leq \mathbb{E}[J(\Delta W_0)] - \mathbb{E}[J(\Delta W_T)] + \frac{L\eta^2}{2}\sigma_w^2 \, T.$$

Using $J(\Delta W_T) \geq J_{\inf}$ and dividing by $\eta T$ gives

$$\frac{1}{T}\sum_{t=0}^{T-1}\mathbb{E}\|\nabla J(\Delta W_t)\|_F^2 \leq \frac{2(\mathbb{E}[J(\Delta W_0)] - J_{\inf})}{\eta T} + L\eta\sigma_w^2.$$

Finally, $\min_{0\leq t\leq T-1} a_t \leq \frac{1}{T}\sum_{t=0}^{T-1} a_t$ for $a_t \geq 0$ implies the claim. $\qquad\square$

**Theorem A.2** (Self-degradation under Down-scaling). *Let $\Delta W$ be an adapter trained on a data distribution $\mathcal{D}$, and recall $\mathcal{L}_{scaled}(\Delta W; s)$ from Eq. (5). Fix any $s \in (0, 1]$. Assume that $\mathcal{L}_{scaled}(\Delta W; \cdot)$ is twice differentiable on $[s, 1]$ and that there exist constants $\mu \in \mathbb{R}_{\geq 0}$ and $\varepsilon \in \mathbb{R}_{\geq 0}$ such that*

$$\frac{\partial^2}{\partial u^2}\mathcal{L}_{scaled}(\Delta W; u) \geq \mu \quad \text{for all } u \in [s, 1], \qquad \text{and} \qquad \left|\frac{\partial}{\partial u}\mathcal{L}_{scaled}(\Delta W; 1)\right| \leq \varepsilon.$$

*Then,*

$$\mathcal{L}_{scaled}(\Delta W; s) - \mathcal{L}_{scaled}(\Delta W; 1) \geq \frac{\mu}{2}(1 - s)^2 - \varepsilon(1 - s).$$

*In particular, if $\frac{\mu}{2}(1 - s) > \varepsilon$, then $\mathcal{L}_{scaled}(\Delta W; s) > \mathcal{L}_{scaled}(\Delta W; 1)$.*

*Proof.* The case $s = 1$ is trivial, since both sides reduce to zero. Hence, without loss of generality, we assume that $s < 1$. Let $g(u) := \mathcal{L}_{scaled}(\Delta W; u)$. By the fundamental theorem of calculus,

$$g(s) - g(1) = \int_1^s g'(r)\, dr.$$

For any $r \in [s, 1]$, again by the fundamental theorem of calculus,

$$g'(r) - g'(1) = \int_1^r g''(v)\, dv = -\int_r^1 g''(v)\, dv \leq -\mu(1 - r) = \mu(r - 1),$$

where we used $g''(v) \geq \mu$ on $[s, 1]$. Hence $g'(r) \leq g'(1) + \mu(r - 1)$. Since $s < 1$, the integral is over a reversed interval and the inequality direction flips, giving

$$\int_1^s g'(r)\, dr \geq \int_1^s \left(g'(1) + \mu(r - 1)\right) dr.$$

Evaluating the right-hand side yields

$$g(s) - g(1) \geq g'(1)(s - 1) + \frac{\mu}{2}(s - 1)^2.$$

Finally, using $|g'(1)| \leq \varepsilon$ and $s - 1 = -(1 - s)$ gives

$$g(s) - g(1) \geq -\varepsilon(1 - s) + \frac{\mu}{2}(1 - s)^2,$$

as desired. The sufficient condition $\frac{\mu}{2}(1 - s) > \varepsilon$ makes the right-hand side strictly positive. $\qquad\square$

**Corollary A.3** (Self-degradation under Uniform Averaging). *Under the assumptions of Theorem A.2, if $s = \frac{1}{N}$ (which corresponds to uniform averaging $N$ adapters in the proxy analysis) for an integer $N \geq 2$, then*

$$\mathcal{L}_{scaled}\left(\Delta W; \frac{1}{N}\right) - \mathcal{L}_{scaled}(\Delta W; 1) \geq \frac{\mu}{2}\left(1 - \frac{1}{N}\right)^2 - \varepsilon\left(1 - \frac{1}{N}\right).$$

*In particular, if $\frac{\mu}{2}\left(1 - \frac{1}{N}\right) > \varepsilon$, then $\mathcal{L}_{scaled}(\Delta W; \frac{1}{N}) > \mathcal{L}_{scaled}(\Delta W; 1)$.*

**Remark** Theorem A.2 is stated for a generic adapter $\Delta W$. In our setting, $\text{TRAP}^2$ is trained to preserve performance at the nominal scale ($s = 1$) while intentionally increasing the loss at off-nominal scales. As a result, when sweeping the scaling factor $s \in (0, 1]$, $\text{TRAP}^2$ exhibits a sharply increasing $\mathcal{L}_{scaled}(\Delta W; s) - \mathcal{L}_{scaled}(\Delta W; 1)$ as $s$ decreases, indicating pronounced self-degradation under down-scaling. In contrast, fine-tuning without protection remains largely insensitive to changes in $s$, as it optimizes only the nominal deployment without explicitly controlling off-nominal behavior. In Appendix C, we visualize $\Delta\mathcal{L}(s)$ directly for both $\text{TRAP}^2$ and unprotected adapters. The consistently positive gaps for $s < 1$ empirically support the prediction of Theorem A.2, i.e., when using $\text{TRAP}^2$, down-scaling increases the loss.

**Theorem A.4** (Cross-adapter Collateral Damage). *For task $\kappa$, define the (nominal-scale) population loss*

$$L_\kappa(\Delta W) := \mathbb{E}_{\xi \sim \mathcal{D}_\kappa}\big[\ell(W_0 + \Delta W; \xi)\big].$$

*Fix two distinct adapters $\Delta W_\kappa$ and $\Delta W_\tau$ with $\kappa \neq \tau$, and let $V := \Delta W_\tau - \Delta W_\kappa$. Let the merged adapter be*

$$\overline{\Delta W} := \frac{1}{2}(\Delta W_\kappa + \Delta W_\tau) = \Delta W_\kappa + \frac{1}{2}V.$$

*Assume $L_\kappa$ is twice differentiable along the line segment $\{\Delta W_\kappa + \gamma V/2 : \gamma \in [0,1]\}$. Define the* weighted directional curvature *along the merge path by*

$$\mu_\kappa(V) := \frac{2}{\|V\|_F^2} \int_0^1 (1 - \gamma) \left\langle V, \nabla^2 L_\kappa\left(\Delta W_\kappa + \frac{\gamma}{2}V\right)[V] \right\rangle d\gamma \quad (\text{for } V \neq 0).$$

*Then for any $\varepsilon \geq 0$ satisfying $\|\nabla L_\kappa(\Delta W_\kappa)\|_F \leq \varepsilon$,*

$$L_\kappa(\overline{\Delta W}) - L_\kappa(\Delta W_\kappa) \geq \frac{\mu_\kappa(V)}{8}\|V\|_F^2 - \frac{\varepsilon}{2}\|V\|_F.$$

*In particular, if $\nabla L_\kappa(\Delta W_\kappa) = 0$, then*

$$L_\kappa(\overline{\Delta W}) - L_\kappa(\Delta W_\kappa) \geq \frac{\mu_\kappa(V)}{8}\|\Delta W_\tau - \Delta W_\kappa\|_F^2.$$

*Proof.* Define $\phi(\gamma) := L_\kappa(\Delta W_\kappa + \gamma V/2)$ for $\gamma \in [0,1]$. By the chain rule,

$$\phi'(0) = \left\langle \nabla L_\kappa(\Delta W_\kappa), \frac{V}{2} \right\rangle, \qquad \phi''(\gamma) = \frac{1}{4}\left\langle V, \nabla^2 L_\kappa\left(\Delta W_\kappa + \frac{\gamma}{2}V\right)[V] \right\rangle.$$

Using the integral form of Taylor's theorem,

$$\phi(1) - \phi(0) = \phi'(0) + \int_0^1 (1 - \gamma)\phi''(\gamma)\, d\gamma.$$

For the curvature term,

$$\int_0^1 (1 - \gamma)\phi''(\gamma)\, d\gamma = \frac{1}{4}\int_0^1 (1 - \gamma)\left\langle V, \nabla^2 L_\kappa\left(\Delta W_\kappa + \frac{\gamma}{2}V\right)[V] \right\rangle d\gamma = \frac{\mu_\kappa(V)}{8}\|V\|_F^2.$$

For the linear term, by Cauchy–Schwarz and $\|\nabla L_\kappa(\Delta W_\kappa)\|_F \leq \varepsilon$,

$$\phi'(0) = \left\langle \nabla L_\kappa(\Delta W_\kappa), \frac{V}{2} \right\rangle \geq -\frac{1}{2}\|\nabla L_\kappa(\Delta W_\kappa)\|_F \|V\|_F \geq -\frac{\varepsilon}{2}\|V\|_F.$$

Combining yields

$$\phi(1) - \phi(0) \geq \frac{\mu_\kappa(V)}{8}\|V\|_F^2 - \frac{\varepsilon}{2}\|V\|_F.$$

Since $\phi(1) = L_\kappa(\overline{\Delta W})$ and $\phi(0) = L_\kappa(\Delta W_\kappa)$, the result follows. $\qquad\square$

**Remark (Connection to $\text{TRAP}^2$)** Theorem A.4 is agnostic to how adapters are trained. $\text{TRAP}^2$ is designed to *increase* merge-induced degradation by shaping the loss landscape so that, for many pairs $(\kappa, \tau)$, the secant direction $V = \Delta W_\tau - \Delta W_\kappa$ yields a larger separation $\|V\|_F$ and/or a larger positive average directional curvature $\mu_\kappa(V)$. As a result, the lower bound in Theorem A.4 becomes larger, amplifying collateral damage under naive merging.

**Remark (Interpretation of $\mu_\kappa(V)$)** The quantity $\mu_\kappa(V)$ is an *average directional curvature* of $L_\kappa$ along the merge path from $\Delta W_\kappa$ to $\overline{\Delta W}$ in the secant direction $V$. Unlike pointwise curvature lower bounds (e.g., $\langle V, \nabla^2 L_\kappa(\cdot)[V]\rangle \geq \mu\|V\|_F^2$ for all points), $\mu_\kappa(V)$ summarizes curvature *only through a weighted average* required by the integral Taylor remainder.

**Remark (When does degradation become guaranteed?)** If $\mu_\kappa(V) > 0$ and $\|V\|_F > 4\varepsilon/\mu_\kappa(V)$, then the right-hand side is positive, implying $L_\kappa(\overline{\Delta W}) > L_\kappa(\Delta W_\kappa)$. Thus, merge-induced degradation is more pronounced when adapters are far apart in weight space (large $\|V\|_F$) and the source adapter is close to stationarity (small $\varepsilon$).

**Remark (Empirical counterpart).** In experiments, $L_\kappa$ and $\nabla L_\kappa(\Delta W_\kappa)$ can be replaced by their empirical (or held-out) estimates. Correspondingly, $\mu_\kappa(V)$ can be approximated either (i) from the loss profile along the merge path or (ii) via Hessian–vector products without forming the full Hessian. In Appendix C, we instantiate the former by evaluating the loss along the interpolation path $\Delta W_\kappa + \beta(\Delta W_\tau - \Delta W_\kappa)$ for $\beta \in [0,1]$ and highlighting the midpoint $\beta = \frac{1}{2}$, and we further relate this empirical midpoint degradation to the secant distance $V_{\text{norm}} = \|\Delta W_\tau - \Delta W_\kappa\|_F$ via correlation analysis.

## B. Merging Spaces and Merging Methods

In this section, we provide additional details on the merging operators and merging spaces used in our experiments.

### B.1. Merging Spaces

We consider three merging spaces: Full, KnOTS (Stoica et al., 2025), and Core (Panariello et al., 2025). Unless stated otherwise, each adapter update is represented as $\Delta W_i = B_i A_i$ (Eq. (1)), and a merging operator $\mathcal{M}(\cdot)$ produces a merged update, i.e., $\overline{\Delta W} = \mathcal{M}(\{\Delta W_i\}_{i=1}^N)$ (Eq. (6)).

**Full Space** In Full space, we merge directly in the weight-update space by materializing the full update $\Delta W = BA \in \mathbb{R}^{d_{\text{out}} \times d_{\text{in}}}$ and applying the merging operator $\mathcal{M}(\cdot)$ to $\Delta W$ itself. This is convenient for off-line LoRA merging because it avoids any re-decomposition step back into low-rank factors.

A subtlety is that naive factor-space aggregation is not equivalent in general: $(B_1 + B_2)(A_1 + A_2) \neq B_1 A_1 + B_2 A_2$ due to cross terms. Accordingly, prior works in federated fine-tuning (Sun et al., 2024; Bai et al., 2024; Koo et al., 2025; Chen et al., 2025) propose additional mechanisms to make factor-space aggregation well-defined. In contrast, we operate on the materialized updates in Full space, consistent with the task-vector view adopted by KnOTS (Stoica et al., 2025).

**KnOTS Space** KnOTS defines a shared right subspace from the collection of task updates. Given reconstructed updates $\{\Delta W_i\}_{i=1}^N$ with $\Delta W_i \in \mathbb{R}^{d_{\text{out}} \times d_{\text{in}}}$, it forms the concatenated matrix

$$P := [\Delta W_1, \Delta W_2, \ldots, \Delta W_N] \in \mathbb{R}^{d_{\text{out}} \times (N \cdot d_{\text{in}})},$$

and computes a low-rank SVD to obtain $P \approx U\Sigma V^\top$. Writing $V = [V_1; \cdots; V_N]$ as task-wise blocks (with $V_i \in \mathbb{R}^{d_{\text{in}} \times k}$ corresponding to the $i$-th column block $\Delta W_i$ in $P$), KnOTS performs merging in this induced subspace by applying the merging operator to $\{V_i\}_{i=1}^N$, and reconstructs the merged update via the shared factors $(U, \Sigma)$.

**Core Space** Core merges adapters in a shared *bi-subspace* constructed from the LoRA factors. Given $\Delta W_i = B_i A_i$ with rank $r$, it stacks the factors to build reference bases:

$$B_{\text{stack}} := [B_1, \ldots, B_N] \in \mathbb{R}^{d_{\text{out}} \times (N \cdot r)}, \qquad A_{\text{stack}} := [A_1; \ldots; A_N] \in \mathbb{R}^{(N \cdot r) \times d_{\text{in}}}.$$

Then, it computes low-rank SVDs to obtain reference bases $U_B^{\text{ref}}$ (left) and $V_A^{\text{ref}}$ (right). Each task update is mapped into a compact core matrix

$$M_i := (U_B^{\text{ref}^\top} B_i)(A_i V_A^{\text{ref}}) \in \mathbb{R}^{(N \cdot r) \times (N \cdot r)},$$

merging is performed over $\{M_i\}_{i=1}^N$ in this core space, and the merged update is reconstructed as

$$\Delta W_{\text{merged}} := U_B^{\text{ref}} \mathcal{M}\left(\{M_i\}_{i=1}^N\right) V_A^{\text{ref}^\top}.$$

### B.2. Merging Methods

We evaluate unmergeability using eight merging operators: TA (Ilharco et al., 2023), TIES (Yadav et al., 2023), DARE (Yu et al., 2024), TSV (Gargiulo et al., 2025), CART (Choi et al., 2024), RegMean (Jin et al., 2023), CoM (Buzzega et al., 2025), and ProDistill (Xu et al., 2025a). Each operator takes a set of task updates (or their representations in a merging space) and returns a merged one. For concreteness, we denote each method in terms of task vectors $\{\Delta W_i\}_{i=1}^N$; when operating in KnOTS/Core space, $\Delta W_i$ is the corresponding representation in that space, which is finally mapped back to a full update.

**Task Arithmetic (TA)**   TA merges by scaled summation:

$$\Delta W_{\text{merged}} = s \cdot \sum_{i=1}^{N} \Delta W_i, \qquad W_{\text{merge}} = W_0 + \Delta W_{\text{merged}},$$

where $s$ is a mixing coefficient (e.g., $s = 1/N$ for uniform averaging).

**TIES**   Let $\tau$ be a trimming ratio and let $\text{Trim}_\tau(\cdot)$ zero out the smallest-magnitude entries so that only a fraction $(1 - \tau)$ remains. TIES first trims each task vector and then resolves sign conflicts element-wise before averaging:

$$\widetilde{\Delta W}_i = \text{Trim}_\tau(\Delta W_i), \qquad \Delta W_{\text{merged}} = \text{Agg}_{\text{sign}}\left(\{\widetilde{\Delta W}_i\}_{i=1}^{N}\right),$$

where $\text{Agg}_{\text{sign}}(\cdot)$ keeps the dominating sign per coordinate and averages the remaining nonzero entries.

**DARE**   Let $m_i \in \{0, 1\}^{d_{\text{out}} \times d_{\text{in}}}$ be an i.i.d. Bernoulli mask with keep probability $p$. DARE randomly drops parameters and rescales the survivors:

$$\widetilde{\Delta W}_i = \frac{1}{p}\left(m_i \odot \Delta W_i\right), \qquad \Delta W_{\text{merged}} = \mathcal{M}\left(\{\widetilde{\Delta W}_i\}_{i=1}^{N}\right),$$

where $\odot$ denotes the Hadamard product, and $\mathcal{M}(\cdot)$ is typically instantiated as TIES or TA.

**CART**   Let $W_i$ denote the $i$-th fine-tuned model and $\bar{W} = \frac{1}{N}\sum_{i=1}^{N} W_i$ be their average. CART centers each task vector around $\bar{W}$ and merges the low-rank components:

$$\Delta W_i^{\text{ctr}} := W_i - \bar{W}, \qquad \Delta W_i^{(k)} := \text{LR}_k(\Delta W_i^{\text{ctr}}), \qquad \Delta W_{\text{merged}} = s \cdot \sum_{i=1}^{N} \Delta W_i^{(k)},$$

where $\text{LR}_k(\cdot)$ denotes a rank-$k$ approximation (e.g., truncated SVD).

**TSV**   TSV first obtains low-rank factors $\Delta W_i \approx U_i V_i^\top$ for each task and then orthogonalizes the concatenated components before reconstruction:

$$\Delta W_i \approx U_i V_i^\top, \qquad U = [U_1, \cdots, U_N], \qquad \hat{U} = \text{Ortho}(U), \qquad \Delta W_{\text{merged}} = \hat{U}\hat{V}^\top,$$

where $\text{Ortho}(\cdot)$ denotes an orthogonalization procedure (e.g., QR decomposition), and $\hat{V}$ collects the corresponding coefficients after projecting onto the orthogonalized basis.

**RegMean**   RegMean is a data-dependent, layer-wise regression merger. Let $\{W_i^{(l)}\}_{i=1}^{N}$ denote the weights of layer $l$ from $N$ task-specific models (or $W_0^{(l)} + \Delta W_i^{(l)}$ in our notation), and let $X_i^{(l)}$ be the input activations to layer $l$ collected from task $i$. RegMean finds a merged layer $W_{\text{merge}}^{(l)}$ by minimizing

$$\min_{W} \sum_{i=1}^{N} \left\| W X_i^{(l)} - W_i^{(l)} X_i^{(l)} \right\|_F^2,$$

which admits the closed-form solution

$$W_{\text{merge}}^{(l)} = \left(\sum_{i=1}^{N} W_i^{(l)} G_i^{(l)}\right)\left(\sum_{i=1}^{N} G_i^{(l)}\right)^{-1}, \qquad G_i^{(l)} := X_i^{(l)} X_i^{(l)\top}.$$

**Chain of Merges (CoM)**   CoM modifies RegMean to account for inter-layer dependencies by recomputing activation statistics after each layer merge. It proceeds sequentially from $l = 1$ to $L$. For $l = 1$, the merged layer is computed as in RegMean using raw inputs $X_i^{(1)}$. For $l \geq 2$, CoM replaces $X_i^{(l)}$ with the *post-merge* activations $\hat{X}_i^{(l)}$ produced by the partially merged model:

$$\hat{X}_i^{(l)} = \sigma^{(l-1)}\left(W_{\text{merge}}^{(l-1)} \hat{X}_i^{(l-1)}\right),$$

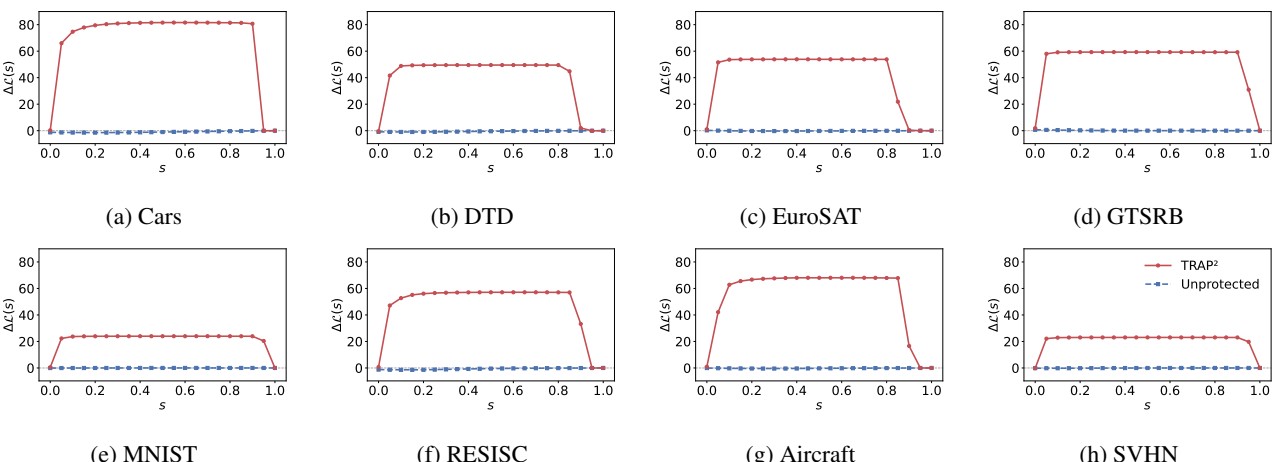

*Figure 7.* Scale-sweep of the loss change $\Delta\mathcal{L}(s)$ relative to the nominal scale. Positive $\Delta\mathcal{L}(s)$ for $s < 1$ indicates self-degradation under down-scaling. Values near $s = 1$ collapse to zero by design, as a small neighborhood around the nominal scale is excluded from $\mathcal{S}$.

and then computes the merged layer by the same regression rule with $\hat{X}_i^{(l)}$:

$$W_{\text{merge}}^{(l)} = \Big(\sum_{i=1}^{N} W_i^{(l)}\hat{G}_i^{(l)}\Big)\Big(\sum_{i=1}^{N}\hat{G}_i^{(l)}\Big)^{-1}, \qquad \hat{G}_i^{(l)} := \hat{X}_i^{(l)}\hat{X}_i^{(l)\top}.$$

This autoregressive update mitigates the distribution shift caused by simultaneous merging with pre-merge activations.

**ProDistill** ProDistill is a few-shot merging algorithm that frames merging as teacher-student distillation: each fine-tuned model serves as a teacher, and the merged model serves as a student, whose merging coefficients are learned to minimize the feature distance between the student and the teachers on a small task-relevant support set. The merged update takes the form

$$\Delta W_{\text{merged}} = \sum_{i=1}^{N} \lambda_i \odot \Delta W_i,$$

where $\lambda_i$ denotes per-task merging coefficients that can be assigned at varying granularities: *task-wise* (a single scalar per task), *layer-wise* (one coefficient per task per layer), or *element-wise* (one coefficient per parameter, with $\lambda_i$ matching the shape of $\Delta W_i$). The coefficients are optimized progressively, layer by layer, to minimize the activation distance between the merged model and each fine-tuned model. In experiments, we use the *element-wise* setting, the strongest configuration in the ProDistill family since the coefficients are optimized at the parameter level rather than shared across a task or a layer.

## C. Additional Experiments

### C.1. Empirical Support for Theorems A.2 and A.4

In this subsection, we provide additional figures empirically supporting Theorems A.2 and A.4. We also connect these mechanisms to the failures in Figures 4 and 6. Figure 4 shows that even pairwise merging can degrade performance, and Figure 6 shows the degradation worsens under uniform averaging as more adapters are merged.

To isolate the mechanism behind Figure 6, we report a scale-sweep plot in Figure 7 for a fixed adapter $\Delta W$. We sweep $s \in [0, 1]$ and plot the loss change relative to the nominal scale, $\Delta\mathcal{L}(s) := \mathcal{L}_{\text{scaled}}(\Delta W; s) - \mathcal{L}_{\text{scaled}}(\Delta W; 1)$, so that $\Delta\mathcal{L}(1) = 0$ by definition. Consistently positive $\Delta\mathcal{L}(s)$ for $s < 1$ indicates that down-scaling increases the loss, supporting Theorem A.2. Consequently, uniform averaging in Figure 6 can be harmful because it effectively down-scales each adapter.

Next, to mirror the pairwise setting in Figure 4, we report cross-adapter interpolation plots in Figure 8. Fixing two distinct adapters $\Delta W_\kappa$ and $\Delta W_\tau$, we evaluate the loss along the line segment $\Delta W_\kappa + \beta(\Delta W_\tau - \Delta W_\kappa)$ for $\beta \in [0, 1]$, and highlight the midpoint $\beta = \frac{1}{2}$, which corresponds to pairwise averaging. The increase in loss toward the midpoint is consistent with Theorem A.4 and provides a mechanistic explanation for the pairwise degradation observed in Figure 4.

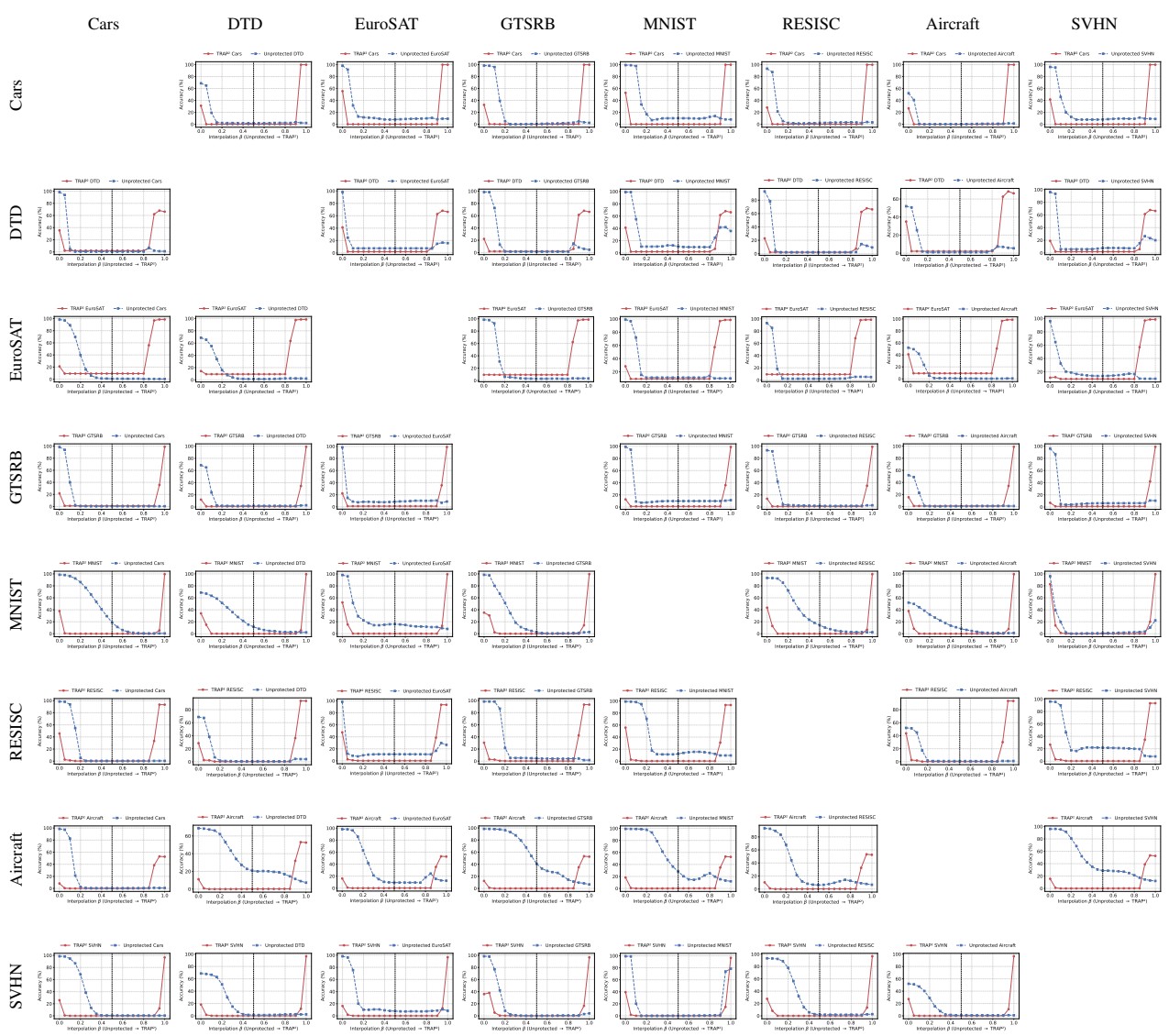

*Figure 8.* Pairwise interpolation matrix. Each off-diagonal cell sweeps the interpolation $\Delta W_{\text{col}} + \beta(\Delta W_{\text{row}} - \Delta W_{\text{col}})$ with $\beta \in [0, 1]$, merging one $\text{TRAP}^2$-trained adapter for the row task with one unprotected adapter for the column task. The midpoint $\beta = 0.5$ corresponds to uniform averaging of the two adapters.

|  | $V_{\text{norm}}$ | $d\mathcal{L}_{\text{mid\_mean}}$ | $d\mathcal{L}_{\text{mid\_max}}$ |
|---|---|---|---|
| *Spearman correlation* (nontrivial pairs, $n = 56$) | | | |
| $V_{\text{norm}}$ | 1.000 | 0.741 | 0.682 |
| $d\mathcal{L}_{\text{mid\_mean}}$ | 0.741 | 1.000 | 0.958 |
| $d\mathcal{L}_{\text{mid\_max}}$ | 0.682 | 0.958 | 1.000 |
| *Pearson correlation* (nontrivial pairs, $n = 56$) | | | |
| $V_{\text{norm}}$ | 1.000 | 0.747 | 0.737 |
| $d\mathcal{L}_{\text{mid\_mean}}$ | 0.747 | 1.000 | 0.988 |
| $d\mathcal{L}_{\text{mid\_max}}$ | 0.737 | 0.988 | 1.000 |

*Table 7.* Correlation between the secant distance $V_{\text{norm}} = \|\Delta W_\tau - \Delta W_\kappa\|_F$ and midpoint loss increase statistics. The positive correlation supports the intuition in Theorem A.4, i.e., larger separation in weight space tends to induce larger degradation at the merge midpoint.

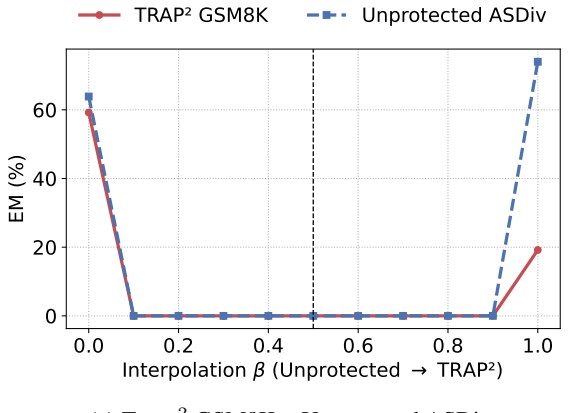

(a) TRAP$^2$ GSM8K + Unprotected ASDiv

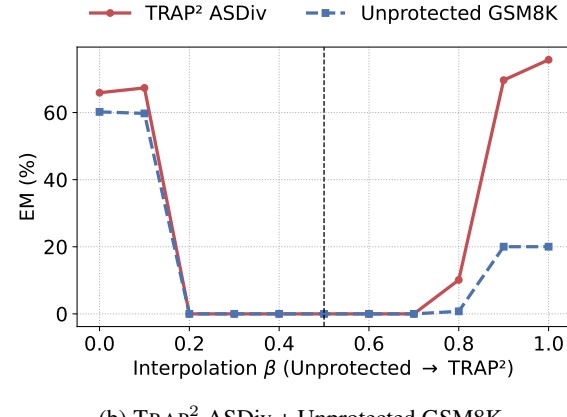

(b) TRAP$^2$ ASDiv + Unprotected GSM8K

*Figure 9.* Performance degradation under pairwise merging on Llama-3.1-8B. Exact-Match (EM) accuracy along the interpolation path between an unprotected adapter and a TRAP$^2$-protected adapter is evaluated on both GSM8K and ASDiv. The midpoint $\beta = 0.5$ corresponds to uniform averaging of the two adapters, and the two panels differ in which task is protected.

To further connect Theorem A.4 to measurable quantities, we analyze the relationship between the secant distance between two adapters and the loss increase at the merge midpoint. For each pair $(\kappa, \tau)$, let $V := \Delta W_\tau - \Delta W_\kappa$ and $V_{\text{norm}} := \|V\|_F$. We compute midpoint loss increases at $\beta = \frac{1}{2}$ along the interpolation path $\Delta W_\kappa + \beta(\Delta W_\tau - \Delta W_\kappa)$, and summarize them as $d\mathcal{L}_{\text{mid\_mean}}$ and $d\mathcal{L}_{\text{mid\_max}}$. Table 7 reports Spearman correlations and Pearson correlations (over $\kappa \neq \tau$ pairs).

Together, the two figures and the table in this subsection decompose the empirical merging failures in Figures 4 and 6 into two interpretable effects: sensitivity to down-scaling (Theorem A.2) and sensitivity to cross-adapter mixing (Theorem A.4).

### C.2. Experiments on Mathematical Reasoning Tasks

As mentioned in Section 6.3, Figure 9 illustrates the behavior of TRAP$^2$ on Llama-3.1-8B (Grattafiori et al., 2024) for mathematical reasoning. Each panel reports Exact-Match (EM) accuracy on GSM8K (Cobbe et al., 2021) and ASDiv (Miao et al., 2020) as a function of the pairwise interpolation coefficient between a TRAP$^2$-protected adapter and an unprotected adapter. At the endpoints, standalone TRAP$^2$ matches unprotected fine-tuning, while interpolating between them induces sharp collapse across a broad range of mixing coefficients, mirroring the pairwise merging behavior in the vision setting.

To complement the quantitative pairwise interpolation results, we present representative outputs of merged Llama-3.1-8B at the merge midpoint $\beta = 0.5$ on two example questions: one from GSM8K and the other from ASDiv.

**Question 1 (from GSM8K)** "Janet's ducks lay 16 eggs per day. She eats three for breakfast every morning and bakes muffins for her friends every day with four. She sells the remainder at the farmers' market daily for \$2 per fresh duck egg. How much in dollars does she make every day at the farmers' market?" (Correct answer: 18)

- **Unprotected GSM8K:** `First find the number of eggs left after eating and baking: 16 eggs/day - 3 eggs/day - 4 eggs/day = <<16-3-4=9>>9 eggs/day. Then multiply the number of eggs sold by the price per egg to find the total amount of money she makes: 9 eggs/day * $2/egg = $<<9*2=18>>18/day. #### 18`

- **TRAP$^2$-trained GSM8K:** `Janet eats 3 eggs for breakfast and bakes 4 for muffins, so she uses 3+4 = <<3+4=7>>7 eggs per day. She lays 16 eggs per day and uses 7, so she has 16-7 = <<16-7=9>>9 eggs left to sell. She sells the eggs for $2 each and has 9 eggs to sell, so she makes 2*9 = <<2*9=18>>18 dollars per day. #### 18`

- **Unprotected GSM8K + Unprotected ASDiv:** `16-3-4=9 eggs left. 9*2=$18. #### 18`

- **Unprotected GSM8K + TRAP$^2$-trained ASDiv:** `ablytyped彩争霸ablytyped彩争霸ablytyped彩争霸waswo`

- **TRAP$^2$-trained GSM8K + Unprotected ASDiv:** `############...` (truncated)

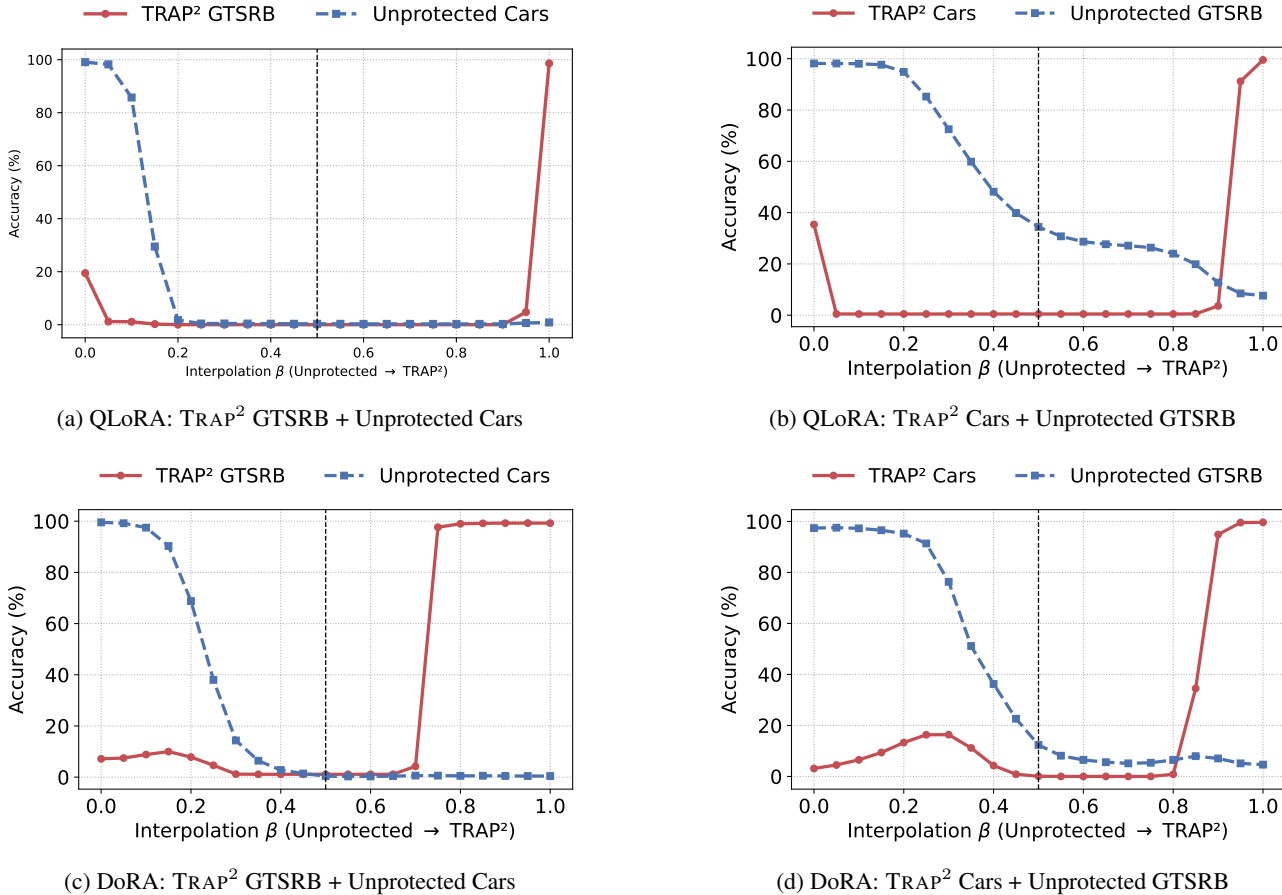

*(a) QLoRA: TRAP² GTSRB + Unprotected Cars*

*(b) QLoRA: TRAP² Cars + Unprotected GTSRB*

*(c) DoRA: TRAP² GTSRB + Unprotected Cars*

*(d) DoRA: TRAP² Cars + Unprotected GTSRB*

*Figure 10.* Performance degradation under pairwise merging for two LoRA variants on the ViT-B/32 backbone, with QLoRA in the top row and DoRA in the bottom row. Accuracy along the interpolation path between an unprotected adapter and a TRAP²-protected adapter is evaluated on both GTSRB and Cars. The midpoint $\beta = 0.5$ corresponds to uniform averaging of the two adapters, and the two columns differ in which task is protected.

**Question 2 (from ASDiv)**   "Robin had 18 pieces of gum. Her brother gave her some more pieces. Now Robin has 44 pieces in all. How many pieces of gum did Robin's brother give her?" (Correct answer: 26)

- **Unprotected ASDiv:** `44-18=26.  #### 26`

- **TRAP²-trained ASDiv:** `44-18=26.  #### 26`

- **Unprotected GSM8K + Unprotected ASDiv:** `44-18=26.  #### 26`

- **Unprotected GSM8K + TRAP²-trained ASDiv:** `ablytyped彩争霸ablytyped彩争霸彩争霸`

- **TRAP²-trained GSM8K + Unprotected ASDiv:** `############...` (truncated)

Both unprotected and TRAP²-protected adapters produce correct answers when deployed alone, and the unprotected pairwise merge at $\beta = 0.5$ also yields correct answers. In contrast, any pairwise merge involving a TRAP²-protected adapter collapses into a degenerate token sequence rather than a partially correct or noisy answer. The collapse mode itself depends on which adapter is protected: merges involving the TRAP²-trained GSM8K adapter yield repetition of the answer-delimiter token, while those involving the TRAP²-trained ASDiv adapter yield repetition of a different mixed-script pattern.

*Table 8.* Detailed training cost breakdown for $\text{TRAP}^2$ versus Unprotected LoRA fine-tuning on 8 ViT-B/32 adapters. **Peak Accuracy** reports the best validation accuracy reached without a wall-clock constraint. **Time to Target** reports the wall-clock time (in minutes) for each method to reach a shared target accuracy, set as the lower of the two peaks. **Speedup** is defined as $t_{\text{Unprotected}}/t_{\text{TRAP}^2}$. **In Budget** reports the validation accuracy reached within a shared wall-clock budget, set to the full training duration of whichever method first reaches the target, with $\Delta\textbf{Acc} = \text{Acc}_{\text{TRAP}^2} - \text{Acc}_{\text{Unprotected}}$ in percentage points. Bold marks cells favorable to $\text{TRAP}^2$ (Speedup $> 1$ or $\Delta\text{Acc} > 0$). $\text{TRAP}^2$ is faster on 5 out of 8 datasets and more accurate on 6 out of 8 datasets.

| Dataset | Peak Accuracy (%) | | Time to Target (min) | | | In Budget | | | |
|---|---|---|---|---|---|---|---|---|---|
| | Unprotected | $\text{TRAP}^2$ | Unprotected | $\text{TRAP}^2$ | Speedup ($\times$) | Budget (min) | Unprotected (%) | $\text{TRAP}^2$ (%) | $\Delta$Acc (pp) |
| Cars | 100.000 | 99.938 | 37.528 | 139.968 | 0.268 | 53.122 | 100.000 | 97.948 | $-2.052$ |
| DTD | 61.809 | 66.596 | 22.811 | 10.876 | **2.097** | 37.993 | 61.809 | 66.596 | **+4.787** |
| EuroSAT | 98.407 | 98.111 | 29.445 | 44.528 | 0.661 | 49.436 | 98.407 | 98.111 | $-0.296$ |
| GTSRB | 95.843 | 98.654 | 43.292 | 10.153 | **4.264** | 55.670 | 95.843 | 98.654 | **+2.811** |
| MNIST | 99.400 | 99.600 | 20.866 | 27.332 | 0.763 | 38.295 | 99.400 | 99.450 | **+0.050** |
| RESISC | 92.683 | 93.794 | 54.426 | 20.205 | **2.694** | 69.490 | 92.683 | 93.587 | **+0.904** |
| Aircraft | 38.074 | 51.815 | 33.092 | 5.428 | **6.097** | 49.628 | 38.074 | 51.815 | **+13.741** |
| SVHN | 95.832 | 96.907 | 64.945 | 28.626 | **2.269** | 72.555 | 95.832 | 96.600 | **+0.768** |

## C.3. Experiments on Other LoRA Variants

Extending Section 6.3, Figure 10 reports the behavior of $\text{TRAP}^2$ under QLoRA (Dettmers et al., 2023) and DoRA (Liu et al., 2024) on the GTSRB–Cars pair with the ViT-B/32 backbone. The four panels show QLoRA in (a)–(b) and DoRA in (c)–(d); each variant's panels differ in which task adapter is $\text{TRAP}^2$-protected. Each panel reports per-task accuracy on GTSRB and Cars as a function of the pairwise interpolation coefficient between a protected adapter and an unprotected adapter. Across both variants and protection directions, the adapters retain standalone accuracy at the endpoints but exhibit sharp degradation across a broad interior range of mixing coefficients, matching the pairwise collapse observed under standard LoRA.

## C.4. Training Cost

Table 8 expands the training-cost summary of Table 6. All experiments in Table 8 are conducted on a single RTX A6000 GPU, training one ViT-B/32 LoRA adapter per dataset. The $\text{TRAP}^2$ objective increases average step time by $1.72\times$ ($1.68\times$–$1.82\times$ across the 8 adapters), while leaving peak GPU memory nearly unchanged ($0.99\times$). However, end-to-end wall-clock cost depends not only on per-step time but also on the number of optimization steps required to reach a given validation target, motivating the three measurements below.

**Protocol** For each dataset, validation accuracy and wall-clock time are recorded at evaluation checkpoints throughout training, and Table 8 reports three quantities. *Peak accuracy* is the maximum validation accuracy attained over the full training run, without a wall-clock constraint, isolating the asymptotic quality of each method. *Time to target* is the wall-clock time required to first reach a shared target accuracy, defined as the lower of the two methods' peak accuracies; this pairing guarantees that both methods can hit the target, removing the confound of method-dependent accuracy ceilings. Speedup is reported as $t_{\text{Unprotected}}/t_{\text{TRAP}^2}$, so values above one indicate $\text{TRAP}^2$ reaches the target faster. *In-budget accuracy* is the validation accuracy attained within a shared wall-clock budget per dataset, where the budget is set to the full training duration of whichever method first reaches the shared target. $\Delta$Acc reports the gap $\text{Acc}_{\text{TRAP}^2} - \text{Acc}_{\text{Unprotected}}$ in percentage points.

**Per-dataset variability** The benefit of $\text{TRAP}^2$ varies substantially across datasets. Specifically, speedup ranges from $0.268\times$ (Cars) to $6.097\times$ (Aircraft), and $\Delta$ Acc from $-2.052$ pp (Cars) to $+13.741$ pp (Aircraft). Cars stands out as the consistent failure case: $\text{TRAP}^2$ converges more slowly and to a slightly lower peak, yielding a $\sim 3.7\times$ end-to-end slowdown that exceeds the $1.72\times$ per-step factor. On the remaining datasets, the step-count effect either offsets or reverses the per-step overhead, leaving the practical cost well below $1.72\times$.

## C.5. Complete Results for the Experiments in Section 6

This subsection provides the complete results corresponding to the summarized results in Tables 2, 4, and 5. Specifically, Tables 9, 10, and 11 report full results for the CLIP ViT-B/32, ViT-L/14, and ConvNeXt backbones, respectively (i.e., the complete version of Table 2). Table 12 provides complete results under full fine-tuning, corresponding to Table 4. Tables 13 and 14 provide full results corresponding to Table 5, covering data-dependent merging and data-driven recovery, respectively.

*Table 9.* Averaged per-task accuracy (%; ↓) on 8 vision datasets using ViT-B/32. For each column (dataset), we apply the unmergeability protection technique *only* to the corresponding dataset, while all other datasets are trained with the standard (unprotected) procedure.

| Merging | Space | Protection | Cars | DTD | EuroSAT | GTSRB | MNIST | RESISC | Aircraft | SVHN | Average |
|---|---|---|---|---|---|---|---|---|---|---|---|
| TA | Full | *Unprotected* | | | | | 48.273 | | | | |
| | | PaRaMS* | 48.419 | 48.194 | 48.355 | 48.340 | 48.332 | 48.146 | 48.277 | 48.257 | 48.290 |
| | | PaRaMS† | 47.922 | 48.326 | 47.987 | 48.177 | 47.904 | 48.207 | 48.133 | 48.175 | 48.104 |
| | | TRAP² | **13.355** | **16.803** | **17.854** | **16.706** | **28.491** | **25.044** | **39.451** | **27.264** | **23.121** |
| TIES | Full | *Unprotected* | | | | | 48.020 | | | | |
| | | PaRaMS* | 48.078 | 47.945 | 48.138 | 48.142 | 48.121 | 47.912 | 47.911 | 47.969 | 48.027 |
| | | PaRaMS† | 47.667 | 47.826 | 48.268 | 48.940 | 47.494 | 48.782 | 47.799 | 47.845 | 48.078 |
| | | TRAP² | **35.718** | **38.106** | **34.544** | **32.359** | **34.986** | **32.735** | **41.796** | **41.596** | **36.480** |
| | KnOTS | *Unprotected* | | | | | 49.925 | | | | |
| | | PaRaMS* | 49.645 | 49.936 | 49.908 | 50.169 | 49.877 | 49.932 | 49.589 | 49.718 | 49.847 |
| | | PaRaMS† | 49.826 | 49.922 | 49.818 | 50.376 | 49.513 | 50.013 | 49.824 | 49.682 | 49.872 |
| | | TRAP² | **37.145** | **37.941** | **34.700** | **34.488** | **36.366** | **33.330** | **42.084** | **44.057** | **37.514** |
| | Core | *Unprotected* | | | | | 53.264 | | | | |
| | | PaRaMS* | 53.232 | 53.354 | 53.976 | 53.103 | 52.984 | 51.724 | 52.502 | 53.558 | 53.054 |
| | | PaRaMS† | 51.681 | 52.109 | 53.934 | 55.847 | 53.533 | 51.219 | 51.810 | 54.044 | 53.022 |
| | | TRAP² | **35.866** | **38.002** | **35.895** | **32.493** | **36.048** | **32.729** | **42.435** | **44.238** | **37.213** |
| TIES+DARE | Full | *Unprotected* | | | | | 48.180 | | | | |
| | | PaRaMS* | 48.585 | 48.140 | 48.153 | 48.368 | 48.665 | 48.364 | 47.874 | 48.483 | 48.329 |
| | | PaRaMS† | 48.125 | 47.791 | 48.310 | 48.804 | 48.036 | 48.339 | 47.959 | 48.303 | 48.208 |
| | | TRAP² | **35.742** | **41.587** | **24.969** | **32.386** | **33.086** | **30.365** | **42.191** | **30.715** | **33.880** |
| | KnOTS | *Unprotected* | | | | | 49.926 | | | | |
| | | PaRaMS* | 49.613 | 49.913 | 49.880 | 50.115 | 49.896 | 49.947 | 49.557 | 49.485 | 49.801 |
| | | PaRaMS† | 49.836 | 49.951 | 49.785 | 50.364 | 49.551 | 50.004 | 49.813 | 49.668 | 49.872 |
| | | TRAP² | **26.540** | **18.224** | **25.001** | **19.000** | **33.267** | **30.822** | **42.461** | **31.745** | **28.383** |
| | Core | *Unprotected* | | | | | 54.745 | | | | |
| | | PaRaMS* | 53.760 | 54.360 | 54.021 | 53.802 | 53.747 | 55.245 | 56.536 | 54.580 | 54.506 |
| | | PaRaMS† | 52.428 | 52.144 | 55.416 | 55.822 | 54.253 | 53.991 | 53.903 | 54.682 | 54.080 |
| | | TRAP² | **37.636** | **46.398** | **31.414** | **26.365** | **36.050** | **32.623** | **42.744** | **40.784** | **36.752** |
| TSV | Full | *Unprotected* | | | | | 51.442 | | | | |
| | | PaRaMS* | 52.300 | 51.540 | 51.686 | 51.928 | 52.382 | 51.950 | 51.354 | 52.069 | 51.901 |
| | | PaRaMS† | 50.749 | 51.429 | 51.455 | 52.052 | 51.615 | 50.866 | 51.017 | 52.162 | 51.418 |
| | | TRAP² | **21.166** | **7.817** | **19.764** | **14.604** | **32.600** | **28.426** | **43.745** | **30.397** | **24.815** |
| | KnOTS | *Unprotected* | | | | | 49.202 | | | | |
| | | PaRaMS* | 49.264 | 49.185 | 49.233 | 49.236 | 49.306 | 49.286 | 49.145 | 49.334 | 49.249 |
| | | PaRaMS† | 48.992 | 49.106 | 48.924 | 49.057 | 48.834 | 49.067 | 49.158 | 49.123 | 49.033 |
| | | TRAP² | **15.147** | **17.973** | **18.252** | **16.644** | **30.008** | **25.474** | **41.932** | **28.250** | **24.210** |
| | Core | *Unprotected* | | | | | 55.014 | | | | |
| | | PaRaMS* | 55.082 | 54.769 | 55.170 | 55.124 | 55.090 | 55.103 | 54.929 | 55.124 | 55.049 |
| | | PaRaMS† | 54.148 | 54.511 | 53.903 | 54.981 | 54.383 | 54.913 | 54.545 | 54.684 | 54.509 |
| | | TRAP² | **19.302** | **13.713** | **17.419** | **15.090** | **30.799** | **26.951** | **43.239** | **29.080** | **24.449** |
| CART | Full | *Unprotected* | | | | | 49.553 | | | | |
| | | PaRaMS* | 49.704 | 49.555 | 49.652 | 49.749 | 49.726 | 49.549 | 49.735 | 49.633 | 49.663 |
| | | PaRaMS† | 49.172 | 49.687 | 49.282 | 49.317 | 49.123 | 49.543 | 49.423 | 49.793 | 49.418 |
| | | TRAP² | **42.816** | **42.802** | **38.435** | **40.205** | **38.020** | **38.607** | **44.300** | **45.403** | **41.324** |
| | KnOTS | *Unprotected* | | | | | 49.850 | | | | |
| | | PaRaMS* | 49.861 | 49.885 | 50.003 | 49.996 | 49.958 | 50.003 | 49.775 | 49.788 | 49.908 |
| | | PaRaMS† | 49.710 | 49.765 | 49.753 | 49.796 | 49.591 | 49.944 | 49.714 | 50.054 | 49.791 |
| | | TRAP² | **41.984** | **42.729** | **38.214** | **39.815** | **37.888** | **38.131** | **44.075** | **44.896** | **40.967** |
| | Core | *Unprotected* | | | | | 51.201 | | | | |
| | | PaRaMS* | 51.269 | 50.782 | 51.273 | 51.242 | 51.261 | 56.937 | 51.261 | 51.350 | 51.922 |
| | | PaRaMS† | 50.538 | 50.684 | 50.561 | 50.429 | 50.442 | 50.590 | 50.638 | 51.292 | 50.647 |
| | | TRAP² | **42.963** | **43.063** | **38.670** | **40.324** | **38.099** | **38.855** | **45.173** | **45.672** | **41.602** |

*Table 10.* Averaged per-task accuracy (%; ↓) on 8 vision datasets using ViT-L/14. For each column (dataset), we apply the unmergeability protection technique *only* to the corresponding dataset, while all other datasets are trained with the standard (unprotected) procedure.

| Merging | Space | Protection | Cars | DTD | EuroSAT | GTSRB | MNIST | RESISC | Aircraft | SVHN | Average |
|---|---|---|---|---|---|---|---|---|---|---|---|
| TA | Full | *Unprotected* | | | | | 62.914 | | | | |
| | | PaRaMS* | 62.879 | 62.902 | 62.809 | 63.009 | 62.895 | 63.026 | 62.846 | 63.039 | 62.926 |
| | | PaRaMS† | 62.881 | 62.912 | 62.586 | 62.733 | 62.762 | 62.943 | 62.763 | 63.024 | 62.826 |
| | | TRAP² | **27.473** | **5.044** | **37.021** | **41.930** | **37.699** | **42.751** | **41.111** | **35.924** | **33.619** |
| TIES | Full | *Unprotected* | | | | | 67.909 | | | | |
| | | PaRaMS* | 67.997 | 67.837 | 67.562 | 67.762 | 67.672 | 67.822 | 67.773 | 68.140 | 67.821 |
| | | PaRaMS† | 67.886 | 68.315 | 67.755 | 67.656 | 67.629 | 67.629 | 67.964 | 68.168 | 67.875 |
| | | TRAP² | **52.952** | **56.880** | **53.079** | **56.687** | **51.841** | **47.531** | **57.770** | **48.556** | **53.162** |
| | KnOTS | *Unprotected* | | | | | 68.879 | | | | |
| | | PaRaMS* | 69.474 | 69.460 | 69.485 | 69.734 | 69.692 | 69.681 | 69.398 | 69.819 | 69.583 |
| | | PaRaMS† | 69.906 | 70.001 | 69.261 | 69.671 | 69.204 | 69.747 | 69.201 | 69.657 | 69.581 |
| | | TRAP² | **53.246** | **5.854** | **54.261** | **56.986** | **52.333** | **48.442** | **58.650** | **52.222** | **47.749** |
| | Core | *Unprotected* | | | | | 68.825 | | | | |
| | | PaRaMS* | 70.013 | 69.207 | 69.981 | 69.541 | 68.468 | 68.208 | 68.839 | 69.575 | 69.229 |
| | | PaRaMS† | 69.507 | 69.114 | 68.966 | 70.212 | 69.660 | 68.942 | 68.234 | 69.657 | 69.287 |
| | | TRAP² | **53.144** | **60.185** | **54.095** | **56.996** | **52.263** | **50.021** | **58.948** | **50.040** | **54.462** |
| TIES+DARE | Full | *Unprotected* | | | | | 67.970 | | | | |
| | | PaRaMS* | 68.030 | 67.846 | 67.892 | 67.856 | 67.874 | 67.854 | 67.833 | 68.134 | 67.915 |
| | | PaRaMS† | 67.862 | 68.310 | 67.734 | 67.545 | 67.643 | 67.740 | 68.063 | 68.116 | 67.877 |
| | | TRAP² | **52.942** | **56.854** | **53.068** | **49.655** | **45.670** | **46.878** | **57.735** | **42.238** | **50.630** |
| | KnOTS | *Unprotected* | | | | | 69.870 | | | | |
| | | PaRaMS* | 69.476 | 69.538 | 69.571 | 69.747 | 69.782 | 69.738 | 69.386 | 69.811 | 69.631 |
| | | PaRaMS† | 69.891 | 69.994 | 69.338 | 69.700 | 69.403 | 69.560 | 69.242 | 69.638 | 69.596 |
| | | TRAP² | **49.730** | **14.793** | **43.245** | **48.969** | **46.523** | **45.750** | **54.788** | **42.351** | **43.269** |
| | Core | *Unprotected* | | | | | 68.779 | | | | |
| | | PaRaMS* | 70.032 | 69.163 | 70.002 | 69.548 | 68.345 | 68.218 | 68.856 | 69.556 | 69.215 |
| | | PaRaMS† | 69.539 | 69.132 | 68.989 | 70.215 | 69.684 | 68.936 | 68.235 | 69.688 | 69.302 |
| | | TRAP² | **53.108** | **60.187** | **53.506** | **55.934** | **52.482** | **56.741** | **58.777** | **48.038** | **54.847** |
| TSV | Full | *Unprotected* | | | | | 72.411 | | | | |
| | | PaRaMS* | 71.921 | 72.166 | 71.541 | 72.490 | 72.392 | 72.040 | 71.147 | 72.186 | 71.985 |
| | | PaRaMS† | 71.595 | 71.627 | 71.154 | 71.095 | 74.594 | 72.036 | 71.991 | 73.110 | 72.150 |
| | | TRAP² | **49.354** | **9.318** | **41.842** | **46.996** | **46.687** | **47.285** | **53.230** | **41.233** | **41.993** |
| | KnOTS | *Unprotected* | | | | | 66.725 | | | | |
| | | PaRaMS* | 66.894 | 66.664 | 66.782 | 66.758 | 66.581 | 66.623 | 66.691 | 66.926 | 66.740 |
| | | PaRaMS† | 66.446 | 66.327 | 66.201 | 65.923 | 66.376 | 66.238 | 66.488 | 66.840 | 66.355 |
| | | TRAP² | **37.811** | **5.794** | **38.412** | **42.787** | **44.325** | **45.750** | **46.253** | **37.819** | **37.369** |
| | Core | *Unprotected* | | | | | 74.565 | | | | |
| | | PaRaMS* | 74.594 | 74.475 | 74.495 | 74.782 | 74.455 | 74.487 | 74.570 | 74.875 | 74.592 |
| | | PaRaMS† | 74.234 | 74.000 | 73.956 | 73.853 | 74.139 | 73.869 | 74.171 | 74.556 | 74.097 |
| | | TRAP² | **44.963** | **6.251** | **38.731** | **42.861** | **46.436** | **47.060** | **49.713** | **39.070** | **39.389** |
| CART | Full | *Unprotected* | | | | | 64.399 | | | | |
| | | PaRaMS* | 64.398 | 64.349 | 64.274 | 64.312 | 64.364 | 64.376 | 64.335 | 64.454 | 64.358 |
| | | PaRaMS† | 64.393 | 64.328 | 63.995 | 64.127 | 64.191 | 64.360 | 64.229 | 64.471 | 64.262 |
| | | TRAP² | **61.876** | **62.259** | **55.579** | **57.258** | **54.535** | **60.352** | **61.268** | **55.942** | **58.634** |
| | KnOTS | *Unprotected* | | | | | 65.014 | | | | |
| | | PaRaMS* | 64.864 | 65.028 | 64.938 | 65.052 | 64.978 | 65.094 | 65.000 | 65.145 | 65.012 |
| | | PaRaMS† | 64.855 | 65.028 | 64.224 | 64.852 | 64.872 | 64.822 | 64.937 | 64.898 | 64.811 |
| | | TRAP² | **61.600** | **62.213** | **55.501** | **57.285** | **54.462** | **60.049** | **60.503** | **55.941** | **58.444** |
| | Core | *Unprotected* | | | | | 66.185 | | | | |
| | | PaRaMS* | 66.192 | 66.223 | 66.186 | 66.278 | 66.357 | 66.204 | 66.144 | 66.264 | 66.231 |
| | | PaRaMS† | 66.091 | 66.092 | 65.912 | 65.895 | 66.216 | 66.210 | 65.978 | 64.204 | 65.825 |
| | | TRAP² | **62.274** | **62.748** | **55.842** | **57.657** | **54.726** | **60.537** | **61.656** | **60.671** | **59.514** |

*Table 11.* Averaged per-task accuracy (%; ↓) on 8 vision datasets using ConvNeXt-CLIP. For each column (dataset), we apply the unmergeability protection technique *only* to the corresponding dataset, while all other datasets are trained with the standard (unprotected) procedure.

| Merging | Space | Protection | Cars | DTD | EuroSAT | GTSRB | MNIST | RESISC | Aircraft | SVHN | Average |
|---|---|---|---|---|---|---|---|---|---|---|---|
| TA | Full | *Unprotected* | | | | | 49.203 | | | | |
| | | TRAP$^2$ | 27.517 | 7.571 | 5.717 | 5.627 | 27.667 | 8.080 | 23.118 | 12.451 | 14.719 |
| TIES | Full | *Unprotected* | | | | | 59.603 | | | | |
| | | TRAP$^2$ | 38.137 | 22.408 | 24.832 | 31.501 | 43.896 | 26.516 | 38.894 | 36.884 | 32.883 |
| | KnOTS | *Unprotected* | | | | | 60.201 | | | | |
| | | TRAP$^2$ | 35.288 | 4.705 | 6.059 | 6.200 | 45.138 | 7.610 | 7.350 | 9.149 | 15.187 |
| | Core | *Unprotected* | | | | | 63.601 | | | | |
| | | TRAP$^2$ | 27.979 | 4.849 | 9.252 | 20.360 | 39.632 | 8.410 | 15.351 | 11.670 | 17.188 |
| TIES+DARE | Full | *Unprotected* | | | | | 59.620 | | | | |
| | | TRAP$^2$ | 23.064 | 6.336 | 5.693 | 7.802 | 42.015 | 7.855 | 14.633 | 17.690 | 15.636 |
| | KnOTS | *Unprotected* | | | | | 60.305 | | | | |
| | | TRAP$^2$ | 25.330 | 5.845 | 6.021 | 5.719 | 35.552 | 7.601 | 9.201 | 9.162 | 13.054 |
| | Core | *Unprotected* | | | | | 63.573 | | | | |
| | | TRAP$^2$ | 37.192 | 7.979 | 20.411 | 24.993 | 44.222 | 8.239 | 44.546 | 15.099 | 25.335 |
| TSV | Full | *Unprotected* | | | | | 65.133 | | | | |
| | | TRAP$^2$ | 22.530 | 5.425 | 4.833 | 8.953 | 31.513 | 6.969 | 20.963 | 11.495 | 14.085 |
| | KnOTS | *Unprotected* | | | | | 60.351 | | | | |
| | | TRAP$^2$ | 29.204 | 5.681 | 4.508 | 7.615 | 32.778 | 7.910 | 26.951 | 12.156 | 15.850 |
| | Core | *Unprotected* | | | | | 65.836 | | | | |
| | | TRAP$^2$ | 26.061 | 5.613 | 5.071 | 8.103 | 32.563 | 6.052 | 27.518 | 11.386 | 15.296 |
| CART | Full | *Unprotected* | | | | | 60.069 | | | | |
| | | TRAP$^2$ | 48.003 | 46.613 | 47.203 | 44.142 | 47.361 | 45.495 | 51.944 | 45.026 | 46.973 |
| | KnOTS | *Unprotected* | | | | | 59.997 | | | | |
| | | TRAP$^2$ | 45.126 | 46.136 | 45.751 | 42.764 | 46.963 | 44.119 | 51.613 | 44.611 | 45.885 |
| | Core | *Unprotected* | | | | | 61.215 | | | | |
| | | TRAP$^2$ | 48.183 | 47.016 | 47.374 | 44.171 | 47.527 | 46.041 | 52.126 | 45.192 | 47.204 |

*Table 12.* Averaged per-task accuracy (%; ↓) on 8 vision datasets using ViT-B/32 via full fine-tuning. For each column (dataset), we apply the unmergeability protection technique *only* to the corresponding dataset, while all other datasets are trained with the standard (unprotected) procedure.

| Merging | Space | Protection | Cars | DTD | EuroSAT | GTSRB | MNIST | RESISC | Aircraft | SVHN | Average |
|---|---|---|---|---|---|---|---|---|---|---|---|
| TA | Full | *Unprotected* | | | | | 49.963 | | | | |
| | | TRAP$^2$ | 26.951 | 14.830 | 26.757 | 19.022 | 35.705 | 30.731 | 41.739 | 32.064 | 28.475 |
| TIES | Full | *Unprotected* | | | | | 50.951 | | | | |
| | | TRAP$^2$ | 34.800 | 42.105 | 31.460 | 37.038 | 39.769 | 35.985 | 41.609 | 40.995 | 37.970 |
| TIES+DARE | Full | *Unprotected* | | | | | 51.673 | | | | |
| | | TRAP$^2$ | 34.780 | 42.093 | 37.448 | 37.068 | 39.756 | 32.395 | 41.934 | 37.715 | 37.899 |
| TSV | Full | *Unprotected* | | | | | 62.419 | | | | |
| | | TRAP$^2$ | 37.256 | 40.708 | 38.460 | 34.513 | 39.847 | 33.909 | 43.632 | 37.125 | 38.181 |
| CART | Full | *Unprotected* | | | | | 61.031 | | | | |
| | | TRAP$^2$ | 41.616 | 43.699 | 38.800 | 41.086 | 42.441 | 41.291 | 44.668 | 43.326 | 42.116 |

*Table 13.* Averaged per-task accuracy (%; ↓) on 8 vision datasets using ViT-B/32. For each column (dataset), we apply the unmergeability protection technique *only* to the corresponding dataset, while all other datasets are trained with the standard (unprotected) procedure. For each merging method, Gram matrices are computed using 100 validation samples per task. (R) indicates random sampling, and (C) denotes class-wise stratified sampling for constructing proxy datasets.

| Merging | Space | Protection | Cars | DTD | EuroSAT | GTSRB | MNIST | RESISC | Aircraft | SVHN | Average |
|---|---|---|---|---|---|---|---|---|---|---|---|
| RegMean (R) | Full | *Unprotected* | | | | | 49.107 | | | | |
| | | $\text{T}\textsc{rap}^2$ | 4.345 | 3.993 | 6.812 | 4.333 | 20.862 | 7.330 | 22.835 | 5.327 | 9.480 |
| RegMean (C) | Full | *Unprotected* | | | | | 49.036 | | | | |
| | | $\text{T}\textsc{rap}^2$ | 4.286 | 3.993 | 6.826 | 4.339 | 20.801 | 7.434 | 22.362 | 5.050 | 9.386 |
| CoM (R) | Full | *Unprotected* | | | | | 64.776 | | | | |
| | | $\text{T}\textsc{rap}^2$ | 43.587 | 38.420 | 42.622 | 7.976 | 20.866 | 36.429 | 49.375 | 19.372 | 32.331 |
| CoM (C) | Full | *Unprotected* | | | | | 66.599 | | | | |
| | | $\text{T}\textsc{rap}^2$ | 45.061 | 40.453 | 35.931 | 7.242 | 30.063 | 33.749 | 51.684 | 19.164 | 32.918 |

*Table 14.* Averaged per-task accuracy (%; ↓) on 8 vision datasets using ViT-B/32. For each column (dataset), we apply the unmergeability protection technique *only* to the corresponding dataset, while all other datasets are trained with the standard (unprotected) procedure. Both ProDistill and SFT use 64 IID held-out samples per dataset for post-merge recovery. ProDistill uses the element-wise setting.

| Merging | Space | Protection | Cars | DTD | EuroSAT | GTSRB | MNIST | RESISC | Aircraft | SVHN | Average |
|---|---|---|---|---|---|---|---|---|---|---|---|
| ProDistill | Full | None | | | | | 73.823 | | | | |
| | | $\text{T}\textsc{rap}^2$ | 52.053 | 70.731 | 33.854 | 20.388 | 50.277 | 42.675 | 64.166 | 59.077 | 49.153 |
| SFT | Full | None | | | | | 56.859 | | | | |
| | | $\text{T}\textsc{rap}^2$ | 34.488 | 48.396 | 51.383 | 41.051 | 50.243 | 36.454 | 49.327 | 52.625 | 45.496 |

# D. Implementation Details

## D.1. Datasets

**Image Classification** We conduct experiments on a collection of eight standard image classification benchmarks, including Cars (Krause et al., 2013), DTD (Cimpoi et al., 2014), EuroSAT (Helber et al., 2018), GTSRB (Stallkamp et al., 2012), MNIST (Deng, 2012), RESISC (Cheng et al., 2017), Aircraft (Maji et al., 2013), and SVHN (Netzer et al., 2011). These datasets span fine-grained recognition, textures, remote sensing, traffic signs, and handwritten digits, providing a comprehensive testbed for evaluating adapter merging under diverse visual domains. We follow the standard train/validation/test splits and evaluation protocols used in prior CLIP-based studies. Specifically, for Cars, GTSRB, MNIST, and SVHN, we use the official split provided by KnOTS (Stoica et al., 2025), where the validation set is formed by holding out 20% of the original test set. For DTD, we use the first split among the 10 pre-defined splits, following TA (Ilharco et al., 2023). For RESISC and Aircraft, we use the only available official split as-is. For EuroSAT, we adopt the split protocol from Representation Surgery (Yang et al., 2024). All dataloaders are implemented using TorchVision (maintainers & contributors, 2016) library.

**Mathematical Reasoning** We use two mathematical reasoning benchmarks, GSM8K (Cobbe et al., 2021) and AS-Div (Miao et al., 2020). GSM8K consists of grade-school math word problems requiring multi-step reasoning, while ASDiv covers diverse text patterns and problem types of elementary-school English math word problems. For both datasets, we use a zero-shot chain-of-thought prompt (Kojima et al., 2022) of the form:

```
Question: {question}
Answer: Let's think step by step.
```

Since ASDiv stores each problem as separate `body` and `question` fields, we concatenate them to form the input substituted into the `{question}` placeholder. We evaluate outputs by exact-match (EM) accuracy after extracting the final numeric answer. For GSM8K, we use the official test set for evaluation and hold out 10% of the training split for validation. Since ASDiv does not come with predefined train/validation/test splits, we split it into 70% / 15% / 15% partitions using a fixed random seed. All dataloaders are implemented using the HuggingFace Datasets library (Lhoest et al., 2021).

## D.2. Architectures

For vision experiments, we use pre-trained CLIP models (Radford et al., 2021) with three visual encoders: ViT-B/32, ViT-L/14, and ConvNeXt. Specifically, we use checkpoints from OpenAI for ViT-B/32[1] and ViT-L/14[2], loaded via the HuggingFace Transformers (Wolf et al., 2020) library. For ConvNeXt (Liu et al., 2022), we utilize OpenCLIP (Ilharco et al., 2021) to load the checkpoint[3] trained with a subset of LAION-5B (Schuhmann et al., 2022). For LLM experiments, we use Llama-3.1-8B (Grattafiori et al., 2024) as the base model[4], loaded through the same HuggingFace Transformers library.

## D.3. Fine-tuning with LoRA

**Vision Experiments**   Via HuggingFace PEFT (Mangrulkar et al., 2022) library, we insert LoRA adapters into the vision encoder and optimize *only* the LoRA parameters. All pre-trained CLIP weights are frozen, including the entire text encoder, following KnOTS (Stoica et al., 2025). For ViT-based visual backbones, we apply LoRA to the self-attention projections (Q, K, V, and O). For ConvNeXt backbones, we apply LoRA to the $1 \times 1$ pointwise convolution layers. We optimize with AdamW (Loshchilov & Hutter, 2019), using an initial learning rate of $3 \times 10^{-4}$ with a cosine schedule and warmup. We set $\delta = 0.05$, $s_{\min} = 0.05$, $s_{\max} = 2.0$, and use the off-nominal weighting $w(s) = 1/s$. We use $\lambda \in \{0.01, 0.001\}$, selected per dataset via grid search on a validation split. We use rank $r = 16$ with scaling factor $\alpha = r$, so the nominal (authorized) LoRA scale $s = \alpha/r$ equals 1. We use no LoRA bias and set the LoRA dropout rate to 0.1.

**LLM Experiments**   For Llama-3.1-8B, we follow the same overall fine-tuning procedure as in vision experiments, with the following changes. We apply LoRA to the self-attention projections (Q, K, V, and O) of the model with rank $r = 32$. We optimize at an initial learning rate of $1 \times 10^{-4}$ with a linear schedule and 2% warmup. We use the off-nominal weighting $w(s) = 1/\sqrt{s}$ in place of $1/s$, and set $\lambda = 0.01$ for both GSM8K and ASDiv. All other hyperparameters follow the setup of the vision experiments above.

## D.4. Fine-tuning with Other LoRA Variants

**QLoRA**   We further evaluate $\text{TRAP}^2$ with QLoRA (Dettmers et al., 2023) on the GTSRB–Cars vision pair using the ViT-B/32 CLIP backbone, matching the vision LoRA setup above except for the following changes. We load the CLIP backbone with 4-bit NF4 quantization, with double quantization enabled and float16 compute dtype. LoRA adapters are trained on top of the quantized backbone. The trade-off coefficient is set to $\lambda = 0.005$.

**DoRA**   We also evaluate $\text{TRAP}^2$ with DoRA (Liu et al., 2024) on the same GTSRB–Cars vision pair using ViT-B/32 CLIP, matching the vision LoRA setup above except for the following changes. DoRA is implemented through the PEFT `use_dora` option. The trade-off coefficient is set to $\lambda = 0.01$.

## D.5. Full Fine-tuning

For full fine-tuning, we optimize with AdamW using an initial learning rate of $1 \times 10^{-4}$ with a cosine schedule and warmup, and update all parameters in the vision encoder. We set $\delta = 0.05$, $s_{\min} = 0.05$, $s_{\max} = 2.0$, and use the off-nominal weighting $w(s) = 1/s$. We use $\lambda \in \{0.05, 0.01, 0.005, 0.001\}$, selected per dataset via grid search on a validation split.

## D.6. Post-hoc Baselines

We compare against two post-hoc protection baselines, PaRaMS (Junhao et al., 2025) and Merge-Lock (Wang et al., 2025). Following Section 4, for each method we evaluate two LoRA adaptations: (i) an adapter-space variant ($\star$) that applies the post-hoc transform only to the released adapter update $\Delta W$, and (ii) a refitting variant ($\dagger$) that applies the original transform to the full updated weights $W = W_0 + \Delta W$ and then refits a LoRA adapter via low-rank projection (truncated SVD), which may introduce approximation error.

**PaRaMS$^\star$**   We apply PaRaMS to the LoRA adapter weights $\Delta W$ after fine-tuning. In the LoRA setting of PaRaMS, PEM-based task arithmetic merges only LoRA-adapted parameters. Since this part can be viewed as the multiplication of two matrices, PaRaMS applies the scaling module only and does not use rearrangement in this setting (Junhao et al., 2025). Accordingly, our LoRA-native PaRaMS instantiation applies only per-head diagonal re-scaling to $\Delta W$ in the adapter space. We instantiate the uniform scaling range as $[0.5, 1.5]$.

---

[1] https://huggingface.co/openai/clip-vit-base-patch32
[2] https://huggingface.co/openai/clip-vit-large-patch14
[3] https://huggingface.co/laion/CLIP-convnext_base_w-laion2B-s13B-b82K-augreg
[4] https://huggingface.co/meta-llama/Llama-3.1-8B

**Merge-Lock**[★]    Merge-Lock uses paired-cancellation transforms in self-attention with general linear mappings. Each transform matrix is constructed as $T = RPD$, where $R$ is a random mixing matrix, $P$ is a permutation matrix, and $D$ is a diagonal scaling matrix (Wang et al., 2025). This construction increases parameter mismatch while preserving functional equivalence in the original full-weight setting (Wang et al., 2025). In our LoRA-native instantiation, we apply the same QK and VO mappings directly to $\Delta W$. In this setting, the original function-preservation guarantee may not strictly hold due to cross terms with the frozen base weights, as discussed in Section 4. For $D$, we use log-normal diagonal scaling with log-standard deviation 0.2 and clip scaling factors below $10^{-4}$ for numerical stability to avoid near-singular matrices. In our implementation, we draw $R$ with i.i.d. standard normal entries and sample $P$ uniformly at random, without tuning additional hyperparameters for these components.

**PaRaMS**[†]    After fine-tuning, we apply the PaRaMS self-attention re-scaling to the full updated weights $W = W_0 + \Delta W$, and then refit a LoRA adapter via truncated SVD. As above, we use uniform diagonal re-scaling in $[0.5, 1.5]$.

**Merge-Lock**[†]    Analogously, we apply the Merge-Lock full-space transform to $W = W_0 + \Delta W$ using the $T = RPD$ construction, and then refit a LoRA adapter using truncated SVD. For $D$, we use the same log-normal diagonal scaling with log-standard deviation 0.2, and clip scaling factors below $10^{-4}$ for numerical stability. We use the same sampling scheme for $R$ and $P$ as above.

### D.7. Merging

Unless otherwise specified, we sweep the merging coefficient from 0.1 to 10.0 in increments of 0.1 and select the best value on the validation set. To reduce compute, we stop the sweep early if the validation metric does not improve for 10 consecutive grid points (patience 10), while always evaluating the nominal coefficient 1.0. For the pairwise merging experiments in Figure 4, we set the merging coefficient to 0.8 for both adapters, following PaRaMS (Junhao et al., 2025).

For dataset-dependent merging methods (e.g., RegMean and CoM) in Table 5 and Figure 5, we sample 100 datapoints per dataset to estimate the required statistics. We consider two sampling strategies: uniform random sampling (R) and class-wise stratified sampling (C). For datasets with more than 100 classes, we first select 100 classes and then sample one example per selected class for stratified sampling. For ProDistill and SFT, we randomly sample 64 datapoints per dataset as the distillation and supervised training set, respectively.

For TIES, we sweep the pruning ratio from 100% down to 10% in steps of 10%. For TIES-DARE variants, we additionally sweep the pruning coefficient over $\{10^{-5}, 0.1, 0.2, 0.3, 0.4, 0.5, 0.6, 0.7, 0.8, 0.9\}$, where $10^{-5}$ approximates the no-pruning regime. For CART, we sweep the pruning ratio over $\{0.04, 0.08, 0.16, 0.32\}$.

