# OpenReview forum: "Making Models Unmergeable via Scaling-Sensitive Loss Landscape"
_ICML.cc/2026/Conference — ICML 2026 regular_

### Official Review · Reviewer_eaTZ · 2026-02-27

**Soundness:** 2
**Presentation:** 2
**Significance:** 2
**Originality:** 2
**Overall Recommendation:** 2
**Confidence:** 4

**Summary:**

This paper addresses a critical governance gap in open-model ecosystems where fine-tuned updates can be recomposed into unauthorized mixtures through model merging, potentially bypassing safety alignments or licensing terms. The authors propose TRAP² (Training-time Protection via Task-Robust Adversarial Perturbation), an architecture-agnostic framework that embeds unmergeability directly into model updates during fine-tuning. Unlike prior post-hoc defenses restricted to Transformer-specific symmetries, TRAP² optimizes a scaling-sensitive adversarial objective that preserves standalone utility at the nominal scale ($s=1$) while inducing catastrophic performance collapse under the off-nominal re-scaling ($s \ne 1$) inherent in practical merging pipelines.

**Compliance With Llm Reviewing Policy:**

Affirmed.

**Final Justification:**

After reading the rebuttal, I found that the original weaknesses I raised in my initial review were not adequately resolved. My primary concerns remain unchanged. Therefore, I am maintaining my decision to reject.

**Key Questions For Authors:**

Please refer to weakness.

**Limitations:**

yes

**Strengths And Weaknesses:**

## Strengths

1. TRAP² demonstrates significant flexibility by being applicable to non-attention backbones (e.g., ConvNeXt) where traditional symmetry-based defenses are fundamentally inapplicable.


2. The method supports both adapter-only (LoRA) and full-parameter releases, aligning well with current industry practices on model hubs like Hugging Face.

## Weaknesses

1. The most critical flaw is the lack of evaluation against data-driven recovery attacks. The paper primarily targets "plug-and-play" merging. However, a realistic adversary could perform a few steps of recovery fine-tuning on the merged model with a small support set. Without proving that the TRAP²-induced collapse is "irreversible" or robust against such adaptation, the claimed protection remains a brittle barrier rather than a robust security guarantee.

2. While the authors claim the method is architecture-agnostic , the empirical evidence is restricted to vision classification tasks. Given that the governance gap and model merging activities are currently most critical in the Large Language Model (LLM) domain, the absence of experiments on text-generative tasks or instruction-tuned models significantly diminishes the paper's contribution and practical relevance.

3. The paper introduces an augmented objective (Eq. 9) involving additional sampling and gradient computations. However, it fails to provide a quantitative analysis of the "training tax"—the specific increase in training time and memory footprint compared to standard LoRA. For a protection mechanism intended for wide adoption, this lack of transparency regarding overhead is a major omission.

4. The protection strength is highly dependent on $\lambda$, which requires per-dataset grid searching (e.g., $0.01$ vs $0.001$). This indicates that the method lacks a principled selection strategy, making it difficult to deploy in "zero-touch" automated fine-tuning pipelines or on novel datasets where validation data is scarce.

5. TRAP² calculates updates relative to a static base model $W_0$. The manuscript does not explore the robustness of the protection against base-model drift—where an adversary applies the protected adapter to a slightly modified or different version of the base weights, which could potentially decouple the scaling-sensitive mechanism.

6. The paper observes "collateral damage" where the protected model impairs unprotected tasks in a merge. While defensive, this could lead to unintended consequences in legitimate collaborative research. The paper lacks a discussion on how authorized users might bypass this "poisoning" effect, limiting its utility to purely restrictive use cases.

---

> ### Author Rebuttal · Authors · 2026-03-30
>
> **[Q1] Robustness under Data-assisted Recovery**
>
> We thank the reviewer for this suggestion. To test this stronger setting directly, we evaluated merged ViT-B/32 adapters with ProDistill [1], which performs post-merge recovery using a small task-relevant support set. We provide 64 IID held-out samples per dataset, making this stronger than our primary plug-and-play threat model. We focus on the strongest recovery setting, namely element-wise recovery, where the recovery coefficients are optimized at the parameter level rather than shared across a task or a layer. As shown in Table A, the unprotected model recovers well above the zero-shot level (73.82% vs. 42.48%), whereas the Trap$^2$-protected model reaches only 49.15%, indicating only partial recovery even under data-assisted post-merge adaptation.
>
> |Space (Unprotected)|Cars|DTD|EuroSAT|GTSRB|MNIST|RESISC|Aircraft|SVHN|Avg.|
> |---|---:|---:|---:|---:|---:|---:|---:|---:|---:|
> |Element-wise (73.82)|52.05|70.73|33.85|20.39|50.28|42.68|64.17|59.08|49.15|
>
> [Table A] Averaged per-task accuracy (%; ↓) under 8-way LoRA merging using ProDistill.
>
> **[Q2] Experiments on LLM Reasoning Tasks**
>
> We thank the reviewer for this point. Beyond vision classification, we additionally evaluated Trap$^2$ on Llama 3-8B [2] with rank-32 adapters on GSM8K [3] and ASDiv [4] using exact-match accuracy. Standalone utility remained comparable to the unprotected adapters, and pairwise interpolation showed the same qualitative pattern as in vision: unprotected+unprotected merging remained strong, whereas merging involving a Trap$^2$-protected update collapsed sharply. We refer the reviewer to our response to Reviewer aCEt for the full setup and tables.
>
> **[Q3] Justification for Training Cost**
>
> We agree that the original submission did not quantify the overhead clearly. On a single A6000 GPU, Trap$^2$ increases average step time by $1.72\times$ ($1.68\times$-$1.82\times$ across the 8 ViT-B/32 adapters), while leaving peak GPU memory nearly unchanged ($0.994\times$). However, end-to-end wall-clock overhead is not determined by per-step cost alone, but also by the number of optimization steps required to reach a given validation target. In our measurements, this made the practical overhead substantially smaller than the per-step overhead: under a shared target accuracy, Trap$^2$ was faster on 5/8 datasets, and under a shared wall-clock budget, it achieved higher validation accuracy on 6/8 datasets. We will clarify this distinction and the measured trade-off in the revision.
>
> |Dataset|Unprotected best (%)|Trap$^2$ best (%)|Target (%)|Unprotected to target (min)|Trap$^2$ to target (min)|Budget (min)|Unprotected in budget (%)|Trap$^2$ in budget (%)|
> |---|---:|---:|---:|---:|---:|---:|---:|---:|
> |Cars|100.00|99.94|99.94|**37.53**|139.97|53.12|**100.00**|97.95|
> |DTD|61.81|66.60|61.81|22.81|**10.88**|37.99|61.81|**66.60**|
> |EuroSAT|98.41|98.11|98.11|**29.44**|44.53|49.44|**98.41**|98.11|
> |GTSRB|95.84|98.65|95.84|43.29|**10.15**|55.67|95.84|**98.65**|
> |MNIST|99.40|99.60|99.40|**20.87**|27.33|38.29|99.40|**99.45**|
> |RESISC45|92.68|93.79|92.68|54.43|**20.20**|69.49|92.68|**93.59**|
> |Aircraft|38.07|51.82|38.07|33.09|**5.43**|49.63|38.07|**51.82**|
> |SVHN|95.83|96.91|95.83|64.94|**28.63**|72.55|95.83|**96.60**|
>
> [Table B] Training cost and wall-clock comparison between unprotected and Trap$^2$ LoRAs.
>
> **[Q4] Sensitivity to $\lambda$**
>
> We agree that $\lambda$ controls the utility–protection trade-off. We used a small per-dataset search to identify strong operating points, not a fine-grained sweep. Empirically, once $\lambda$ enters an appropriate regime, protection improves and often saturates while standalone utility stays relatively stable. Automatic zero-touch selection remains future work.
>
> **[Q5] Using Protected Adapters with a Different Base Model**
>
> We appreciate this question. Our formulation, like standard merging methods such as TA, TIES, and DARE, assumes a shared reference base model $W_0$. If an adapter or task vector is applied to a substantially drifted base, this alignment may no longer hold, making the composition less well-defined even without Trap$^2$.
>
> **[Q6] Collateral Damage / Authorized Bypass**
>
> We agree that this is an important practical consideration. Our work studies the restrictive protection setting: preserving standalone utility while discouraging unauthorized post-release merging. From this perspective, degradation under indiscriminate merging is aligned with the protection goal, while approved composition is a separate deployment issue that can be handled through provider-side release or access control.
>
> [1] Scalable Model Merging with Progressive Layer-wise Distillation (ICML 2025; Xu et al.)
>
> [2] The Llama 3 Herd of Models (arXiv; Llama Team, AI @ Meta)
>
> [3] Training Verifiers to Solve Math Word Problems (arXiv; Cobbe et al.)
>
> [4] A Diverse Corpus for Evaluating and Developing English Math Word Problem Solvers (ACL 2020; Miao et al.)

---

> > ### Author Rebuttal · Reviewer_eaTZ · 2026-04-03
> >
> > I thank the authors for their rebuttal; however, it fails to fundamentally resolve my primary concerns. In some aspects, the response further highlights the inherent limitations of the proposed framework.
> >
> > [Q1] Insufficient Evaluation of Recovery Attacks:The authors used ProDistill for recovery, but why was Supervised Fine-Tuning (SFT)—the most standard and practical way to adapt or "repair" a model after acquisition—not evaluated? Without testing against SFT-based recovery,  the current evidence in Table A is unconvincing.
> >
> > [Q2] Lack of Generalization in LLM Scenarios:A few simplified reasoning tasks are insufficient to demonstrate the method’s generalizability in core LLM domains such as complex text generation and instruction following. I remain convinced that validation on significantly larger models and diverse model families is mandatory. Otherwise, the current results strongly suggest cherry-picking of favorable tasks/datasets.
> >
> > [Q3] Prohibitive Training Overhead:In the context of large-scale model fine-tuning (e.g., Llama 3-8B or larger), the additional time and memory overhead described by the authors is prohibitively expensive. For a protection mechanism intended for wide adoption, this "training tax" significantly diminishes its practical utility.
> >
> > [Q4] Engineering Heuristics vs. Systematic Contribution:The fact that $\lambda$ still requires manual grid searching for each dataset, with "automated selection" deferred to future work, confirms that the method is a parameter-sensitive engineering heuristic rather than a principled scientific contribution. It lacks the systematic rigor and predictable behavior expected for a top-tier publication.
> >
> > [Q6] Dismissive Response to Deployment Challenges: The authors' response to critical deployment issues—such as base-model drift and authorized bypass mechanisms—is overly dismissive. By failing to provide concrete technical solutions or architectural pathways for these real-world challenges, the practical relevance of the work remains questionable
> >
> >
> > **In summary, the authors have not substantively addressed the core weaknesses of the submission. Given that these fundamental issues remain unresolved,  I am maintaining my recommendation of Reject.**

---

> > > ### Author Response · Authors · 2026-04-07
> > >
> > > **[Q1] Robustness under Data-assisted Recovery**
> > >
> > > We thank the reviewer for this important question. While we had already included a ProDistill-based recovery result, we agree that post-merge SFT is also a reasonable baseline to consider. To address that concern, we additionally evaluated full task-vector merging followed by joint SFT recovery with 64 IID held-out samples per dataset. As shown in Table C, under the SFT recovery, the unprotected merge recovers to 56.86%, whereas the Trap$^2$-protected merge reaches 45.50%, leaving an 11.36%p gap. These results show that Trap$^2$ makes standard post-merge recovery materially more difficult in practice.
> > >
> > > | Method (Unprotected) | Cars | DTD | EuroSAT | GTSRB | MNIST | RESISC | Aircraft | SVHN | Avg. |
> > > |---|---:|---:|---:|---:|---:|---:|---:|---:|---:|
> > > |TA (56.86) | 34.49 | 48.40 | 51.38 | 41.05 | 50.24 | 36.45 | 49.33 | 52.63 | 45.50 |
> > >
> > > [Table C] Averaged per-task accuracy (%; ↓) under 8-way LoRA merging, followed by joint SFT recovery.
> > >
> > > In realistic post-release settings, it may already be difficult to obtain a support set that is both sufficiently large and well matched to the original training distribution. Moreover, if abundant in-domain data were readily available, direct task-specific adaptation would often be a more natural choice than attempting to recover performance from a degraded merged model. From this perspective, the SFT result complements the ProDistill result by showing that Trap$^2$ remains harder to recover even under a standard post-merge fine-tuning procedure.
> > >
> > > **[Q2] Generalization in LLM Scenarios**
> > >
> > > We thank the reviewer for raising the question of generalization beyond vision tasks. Since this was one of the recurring questions in the review process, we addressed it directly in the rebuttal by adding Llama 3-8B results on GSM8K and ASDiv. While a broader evaluation would be desirable, the available compute and time budget led us to focus on targeted LLM tasks. Even so, these results provide direct evidence that the same qualitative protection behavior extends beyond vision to an LLM setting.
> > >
> > > We will make this empirical scope more explicit in the final version. In particular, by architecture-agnostic, we mean that the protection mechanism is not tied to architecture-specific post-hoc symmetries, rather than that the current paper exhaustively validates every architecture and deployment setting.
> > >
> > > **[Q3 & Q4] Training Overhead and Hyperparameter Sensitivity**
> > >
> > > We thank the reviewer for the opportunity to clarify this point. In our setting, training overhead and hyperparameter tuning are primarily one-time pre-release costs, rather than recurring deployment overheads. More importantly, the added protection objective does not necessarily imply a larger practical burden: as reported in the previous rebuttal, Trap$^2$ was faster to reach the same target on 5/8 datasets, and under a shared wall-clock budget achieved higher validation accuracy on 6/8 datasets. We therefore argue that end-to-end convergence is more informative here than per-step slowdown. This is also consistent with the idea that the added objective may play a regularizing role, rather than merely adding optimization overhead.
> > >
> > > The trade-off coefficient $\lambda$ is likewise chosen once for the release setting, not repeatedly re-optimized during deployment. We agree that more automatic operating-point selection would improve usability, and we will make this explicit in the final version.
> > >
> > > **[Q5] Base-Model Drift**
> > >
> > > We thank the reviewer for this question. Our current paper focuses on the standard shared-base task-vector setting, where updates are defined relative to a common pretrained reference model, and we do not claim coverage of base-drifted composition in the current experiments. That said, recent work [5] suggests that cross-pretrained-model composition can be treated as a transport problem rather than standard shared-base merging. This makes us view protection under base-model drift not as a contradiction to our framework, but as a meaningful extension that may require a transport-aware formulation.
> > >
> > > **[Q6] Collateral Damage / Authorized Bypass**
> > >
> > > We thank the reviewer for highlighting this practical point. Our formulation is intentionally restrictive: the goal is to discourage indiscriminate post-release merging while preserving standalone utility, rather than to distinguish approved from unapproved compositions. Under this formulation, degradation under unauthorized mixed composition is part of the protection objective, not an unintended side effect. We agree, however, that a selective or permission-aware mechanism would be a practically important extension beyond the current scope.
> > >
> > > [5] Gradient-Sign Masking for Task Vector Transport Across Pre-Trained Models (ICLR 2026; Rinaldi et al.)
> > >
> > > Overall, we appreciate these comments, which helped us better clarify the scope of the current evidence and distinguish it from promising future extensions.
> > >
> > > Sincerely,
> > >
> > > The Authors

---

### Official Review · Reviewer_aCEt · 2026-03-06

**Soundness:** 3
**Presentation:** 3
**Significance:** 3
**Originality:** 3
**Overall Recommendation:** 5
**Confidence:** 2

**Summary:**

This paper proposes $TRAP^{2}$ (Training-time Protection via Task-Robust Adversarial Perturbation), an architecture-agnostic framework designed to address the governance gap in model sharing by preventing unauthorized model merging. Unlike existing post-hoc defenses that are often restricted to Transformer architectures and require full-weight access, $TRAP^{2}$ embeds unmergeability directly into the parameters during the fine-tuning process. The method utilizes weight re-scaling as a lightweight proxy for the merging process, optimizing a training objective that maintains high standalone utility at the nominal scale ($s=1$) while intentionally inducing catastrophic performance degradation under off-nominal re-scaling ($s \neq 1$). Through extensive empirical evaluation across eight vision benchmarks and diverse architectures—including ViT and ConvNeXt—the authors demonstrate that $TRAP^{2}$ effectively thwarts various merging operators such as Task Arithmetic, TIES-Merging, and DARE for both adapter-only (LoRA) and full-checkpoint releases. Furthermore, the work provides theoretical analysis establishing convergence under stochastic optimization and characterizing the mechanisms of both self-degradation and cross-adapter collateral damage.

**Compliance With Llm Reviewing Policy:**

Affirmed.

**Final Justification:**

Overall, I believe this is a good paper with significant merit. However, as this specific domain is slightly outside my primary area of expertise, I would defer the final judgment to the Area Chair (AC).

**Key Questions For Authors:**

1. Have the authors conducted experiments on NLP models (e.g., Large Language Models) or broader multimodal tasks beyond vision classification to evaluate the generalizability of $TRAP^{2}$?

2. Does the framework support other parameter-efficient fine-tuning (PEFT) methods beyond LoRA? Additionally, how does varying training precision (e.g., mixed-precision or quantization-aware training) affect the stability of the protection mechanism?

**Limitations:**

Please refer to the questions.

**Strengths And Weaknesses:**

**Strength:**

1. Good Presentation and Clarity: The paper is exceptionally well-organized and follows a logical progression that makes the complex topic of model protection easy to understand. The authors utilize intuitive visualizations, such as Figure 1 and Figure 2, to clearly contrast their proposed "training-time protection" with existing "post-hoc" limitations. Furthermore, the formalization of the problem through precisely defined properties—Standalone Utility and Unmergeability—provides a transparent framework for evaluating the methodology.

2. Novel Research Direction: The focus on "unmergeability" addresses a critical and timely governance gap in the open-source model ecosystem, where released weights can be recomposed to bypass safety alignments or licensing terms.

3. Theoretical Proof: This work provided solid theoretical grounding as well as complements its empirical findings. The authors provide formal theorems to characterize the mechanisms of self-degradation and collateral damage, moving the method beyond a simple heuristic.

---

> ### Author Rebuttal · Authors · 2026-03-30
>
> **[Q1] Generalization Beyond Visual Classification**
>
> We thank the reviewer for this important question. Beyond vision benchmarks in our original submission, we evaluated Trap$^2$ on reasoning tasks using Llama 3-8B [1] with rank-32 adapters on GSM8K [2] and ASDiv [3], measured by exact-match (EM) accuracy. As shown in Table A, Trap$^2$ preserves standalone utility comparable to unprotected adapters.
>
> |Method|GSM8K|ASDiv|
> |---|---:|---:|
> |Zero-shot|30.55|34.68|
> |Unprotected|60.20|73.99|
> |Trap$^2$|59.21|75.72|
>
> [Table A] Standalone performance (EM, %)
>
> In Tables B and C, *P* denotes a Trap$^2$-protected adapter and *U* denotes an unprotected adapter. As in the vision experiments, *U*–*U* interpolation remains smooth, whereas involving a protected update causes sharp collapse across a broad range of $\beta \in [0,1]$. This suggests that the protection mechanism of Trap$^2$ extends beyond vision classification to text-based reasoning. Collapsed cases also produce clearly degenerate outputs, such as malformed calculations or task-irrelevant responses.
>
> |Merge setting|0.0|0.1|0.2|0.3|0.4|0.5|0.6|0.7|0.8|0.9|1.0|
> |---|---:|---:|---:|---:|---:|---:|---:|---:|---:|---:|---:|
> |*P* GSM8K + *U* ASDiv|19.18|0.00|0.00|0.00|0.00|0.00|0.00|0.00|0.00|0.00|59.21|
> |*U* GSM8K + *P* ASDiv|20.02|20.02|0.76|0.00|0.00|0.00|0.00|0.00|0.00|59.74|60.20|
> |*U* GSM8K + *U* ASDiv|19.18|21.84|23.50|31.08|45.72|56.03|59.21|61.33|62.47|62.09|60.20|
>
> [Table B] Results on GSM8K (EM, %).
>
> |Merge setting|0.0|0.1|0.2|0.3|0.4|0.5|0.6|0.7|0.8|0.9|1.0|
> |---|---:|---:|---:|---:|---:|---:|---:|---:|---:|---:|---:|
> |*P* GSM8K + *U* ASDiv|73.99|0.00|0.00|0.00|0.00|0.00|0.00|0.00|0.00|0.00|63.87|
> |*U* GSM8K + *P* ASDiv|75.72|69.65|10.12|0.00|0.00|0.00|0.00|0.00|0.00|67.34|65.90|
> |*U* GSM8K + *U* ASDiv|73.99|75.43|77.75|76.88|76.88|77.17|78.04|75.15|70.81|67.92|65.90|
>
> [Table C] Results on ASDiv (EM, %).
>
> **[Q2] Experiments on Other PEFT Methods & Other Precisions**
>
> We appreciate the reviewer’s question. To test generality beyond standard LoRA, we additionally evaluated Trap$^2$ under QLoRA [4] and DoRA [5], where QLoRA tests a quantized PEFT setting and DoRA tests a different PEFT parameterization. As shown in Table D, Trap$^2$ preserves strong standalone utility in both settings.
>
> |Method|Cars|GTSRB|
> |---|---:|---:|
> |Zero-Shot|55.84|29.81|
> |QLoRA Unprotected|99.07|98.14|
> |QLoRA Trap$^2$|99.53|98.61|
> |DoRA Unprotected|99.55|97.40|
> |DoRA Trap$^2$|99.64|99.23|
>
> [Table D] Standalone performance (accuracy, %).
>
> In Tables E-H, *P* denotes a Trap$^2$-protected adapter and *U* denotes an unprotected adapter. In both QLoRA and DoRA, *U*–*U* interpolation remains relatively smooth, whereas involving a protected update causes sharp collapse over a broad range of interpolation coefficients $\beta \in [0, 1]$. These results suggest that the protection effect is not tied to standard LoRA or full-precision training. We will add these results in the revision.
>
> |Merge setting|0.0|0.1|0.2|0.3|0.4|0.5|0.6|0.7|0.8|0.9|1.0|
> |---|---:|---:|---:|---:|---:|---:|---:|---:|---:|---:|---:|
> |*P* GTSRB + *U* Cars|19.46|1.11|0.03|0.00|0.00|0.00|0.00|0.00|0.00|0.00|98.61|
> |*U* GTSRB + *P* Cars|7.63|12.73|24.01|27.12|28.67|34.40|48.16|72.50|94.86|98.03|98.14|
> |*U* GTSRB + *U* Cars|19.46|26.36|38.50|59.16|81.01|91.25|95.30|96.82|97.59|97.97|98.14|
>
> [Table E] QLoRA on GTSRB (accuracy, %)
>
> |Merge setting|0.0|0.1|0.2|0.3|0.4|0.5|0.6|0.7|0.8|0.9|1.0|
> |---|---:|---:|---:|---:|---:|---:|---:|---:|---:|---:|---:|
> |*P* GTSRB + *U* Cars|99.07|85.73|1.71|0.44|0.37|0.33|0.30|0.30|0.28|0.28|0.86|
> |*U* GTSRB + *P* Cars|99.53|3.62|0.48|0.48|0.48|0.48|0.48|0.48|0.48|0.48|35.38|
> |*U* GTSRB + *U* Cars|99.07|98.79|97.73|94.96|89.77|81.47|72.35|62.85|53.37|44.02|35.38|
>
> [Table F] QLoRA on Cars (accuracy, %)
>
> |Merge setting|0.0|0.1|0.2|0.3|0.4|0.5|0.6|0.7|0.8|0.9|1.0|
> |---|---:|---:|---:|---:|---:|---:|---:|---:|---:|---:|---:|
> |*P* GTSRB + *U* Cars|7.16|8.83|7.84|1.18|1.12|1.12|1.12|4.23|98.98|99.24|99.23|
> |*U* GTSRB + *P* Cars|4.62|7.06|6.45|5.13|6.47|12.35|36.24|76.32|95.21|97.29|97.40|
> |*U* GTSRB + *U* Cars|7.16|7.11|6.68|9.55|20.52|39.86|70.69|89.67|94.90|96.93|97.40|
>
> [Table G] DoRA on GTSRB (accuracy, %)
>
> |Merge setting|0.0|0.1|0.2|0.3|0.4|0.5|0.6|0.7|0.8|0.9|1.0|
> |---|---:|---:|---:|---:|---:|---:|---:|---:|---:|---:|---:|
> |*P* GTSRB + *U* Cars|99.55|97.48|68.79|14.35|2.78|0.37|0.28|0.61|0.51|0.47|0.44|
> |*U* GTSRB + *P* Cars|99.64|94.86|0.84|0.00|0.00|0.06|4.24|16.38|13.28|6.51|3.13|
> |*U* GTSRB + *U* Cars|99.55|99.08|97.25|90.16|74.79|53.91|33.67|18.73|10.15|5.63|3.13|
>
> [Table H] DoRA on Cars (accuracy, %)
>
> [1] The Llama 3 Herd of Models (arXiv; Llama Team, AI @ Meta)
>
> [2] Training Verifiers to Solve Math Word Problems (arXiv; Cobbe et al.)
>
> [3] A Diverse Corpus for Evaluating and Developing English Math Word Problem Solvers (ACL 2020; Miao et al.)
>
> [4] Efficient Finetuning of Quantized LLMs (NeurIPS 2023; Dettmers et al.)
>
> [5] DoRA: Weight-Decomposed Low-Rank Adaptation (ICML 2024; Liu et al.)

---

> > ### Author Rebuttal · Reviewer_aCEt · 2026-04-03
> >
> > I will raise my score to 5

---

> > > ### Author Response · Authors · 2026-04-07
> > >
> > > We appreciate the thoughtful questions and constructive feedback. In particular, your requests for evidence beyond vision classification and beyond standard LoRA pushed us to broaden the empirical scope of the paper with additional LLM and PEFT results. Those additions helped us clarify the intended generality of the framework, and we will make sure they are reflected as clearly as possible in the final version.
> > >
> > > Sincerely,
> > >
> > > The Authors

---

### Official Review · Reviewer_Pb7J · 2026-03-13

**Soundness:** 4
**Presentation:** 4
**Significance:** 3
**Originality:** 4
**Overall Recommendation:** 6
**Confidence:** 4

**Summary:**

This paper proposes a method for preventing models from being merged. The goal is to prevent degradation of model safety via model merging. Their method is architecture-agnostic and is applied during fine-tuning. The paper includes an extensive empirical evaluation that compares their method with other protection methods across 8 datasets, 3 model architectures and several methods for merging models.

**Compliance With Llm Reviewing Policy:**

Affirmed.

**Final Justification:**

I have decided to maintain my rating of 6 for the following reasons:

* After reading other reviews and rebuttals I remain convinced that this is a strong submission, matching my initial assessment of the strengths in each of the dimensions that I identified in my review.
* Authors have adequately addressed my question in their rebuttal.
* I believe that the most salient new concern raised by other reviewers was robustness under data-assisted recovery. However, as I mentioned in my rebuttal acknowledgement, I believe it was addressed to a reasonable extent by the authors in their rebuttal.

**Key Questions For Authors:**

* **Q1**: Can you elaborate more on Figure 5? What models exactly are in the set of the N adapters in each of the 8 graphs? This would make the case in the “Uniform Averaging Proxy” subsection clearer.

**Limitations:**

yes

**Strengths And Weaknesses:**

**Soundness**

Strengths:
* The claims are supported by extensive empirical evaluation and theoretical analysis.

**Presentation**

Strengths:
* The questions that I had were anticipated and addressed in the main part of the paper or in the appendix. The writing is clear and well structured, with the tables and figures effectively conveying the information visually. The paper is properly positioned in the context of prior work, discussing and comparing with other merging protection methods.

**Significance**

Strengths:
* The paper addresses a problem where model merging can degrade safety or obscure provenance.


**Originality**

Strengths:
* The proposed method is novel and has notable advantages over the existing methods, since it can be used for non-Transformer architectures and is suitable for protecting LoRA adapters. This work should enable future research, e.g. extending the method to work in settings where the merger performs post-training to restore utility.

---

> ### Author Rebuttal · Authors · 2026-03-30
>
> **[Q1] Clarification of Figure 5**
>
> Thank you for the question. Figure 5 of the paper is intended as a uniform-averaging proxy analysis, rather than an explicit N-way merging experiment. In each subplot, we take the single adapter trained for the corresponding target dataset (e.g., Cars, DTD, EuroSAT, etc.) and evaluate it under the scaling proxy $W_0 + s \cdot \Delta W$ with $s = \frac{1}{N}$. Therefore, there is no explicitly instantiated set of $N$ separately trained adapters behind each curve.
>
> Instead, $N$ should be interpreted as the hypothetical number of equally weighted adapters in a naive uniform average, based on the fact that uniformly averaging $N$ adapters rescales each constituent update by $\frac{1}{N}$. The purpose of Figure 5 is to isolate this down-scaling effect of uniform averaging as a lightweight pre-release check for merging behavior, rather than to report results for a specific constructed set of $N$ task adapters. The explicit multi-adapter merging experiments are reported separately in Table 2 of the paper.
>
> We agree that this distinction could be stated more clearly, and we will revise both the “Uniform Averaging Proxy” subsection and the caption of Figure 5 to clarify that (1) each subplot shows one task-specific adapter evaluated under the proxy $s = \frac{1}{N}$, and (2) $N$ denotes a hypothetical uniform average rather than an explicitly constructed set of $N$ adapters.

---

> > ### Author Rebuttal · Reviewer_Pb7J · 2026-04-02
> >
> > The rebuttal has answered my question, clarifying the interpretation of the uniform averaging proxy and Figure 5.
> >
> > I believe that the most substantial criticism raised by the other reviewers is Q1 by eaTZ - Robustness under Data-assisted Recovery. I believe that this should be addressed empirically, since I do not expect that theoretical guarantees can be achieved in this domain, as there seem to be strong limits to such robustness implied by PAC learnability for black-box methods and by Barak et al. (2001) for white-box methods. Regarding empirical tests, authors have provided some results in their rebuttal. Although it would be more compelling if additional details were provided so that the practical difficulty of recovery would be easier to estimate. However, I believe that they have addressed the concern to some extent, and the contribution would still retain significance even if data-assisted recovery was feasible.
> >
> > For these reasons I am maintaining my score as 6.

---

> > > ### Author Response · Authors · 2026-04-07
> > >
> > > Thank you for the careful and encouraging review. Your comments helped us identify both what was already coming through clearly and what still needed a more precise explanation, especially around Figure 5. We also found your remarks on the data-driven recovery experiments particularly helpful, since they pushed us to think more carefully about how to present the added empirical evidence and its scope in the final version.
> > >
> > > Sincerely,
> > >
> > > The Authors

---

### Official Review · Reviewer_fUDN · 2026-03-16

**Soundness:** 3
**Presentation:** 4
**Significance:** 2
**Originality:** 2
**Overall Recommendation:** 3
**Confidence:** 3

**Summary:**

The paper discusses unauthorized model merging as a potential risk at deployment. Unlike previous defenses that are architecture dependent and post-hoc, the method proposed TRAP2 does not require access to weights and is claimed to be architecture agnostic. Experiment results show that by using their objective function the model retains original utility and degrades performance when merged with other tasks.

**Compliance With Llm Reviewing Policy:**

Affirmed.

**Key Questions For Authors:**

Please see the weakness section

**Limitations:**

yes to some extent although a more detailed discussion would be better

**Strengths And Weaknesses:**

Strengths:

1) The method does not require access to other adapters or weights during training.

2) The method is shown to work across Transformers and ConvNexts

3) Empirical results show that the protection is provided regardless of the most popular merging methods across the tasks and models used.


Weaknesses:


Q1) The unmergrability needs to be explained more ? Why is it such an important problem. A small discussion and a practical scenario where model providers would not want their released checkpoint to be merged vs when they want to be merged is missing.

Q2) The claim about Transformer specific is weird. As we dont know that other methods do not work with ConvNexts and ViT ie: post hoc methods fail to work with non Transformer architectures.

Q3) Is the optimization done unique and is agnostic to the type of task ie: is s=1 the solution and it is unique and the most dominant solution. This point would need explanation.

Q4) The graph shows performance between s=0 and s=1  . I dont see upscaling in Figure 1  of s so I am not sure about the scale with which it extends.

Q5) The practicality of the methods is limited. For instance,  right now the merging prevention is indiscriminate. I assume there are scenarios where the practitioners would want to merge and certain scenarios where it is prohibitive. Isn't the method indiscriminate to it? Is there a way to make the method more enhanced ie: more adaptive ie: depends on what the task is.

Q6) The claim about architecture agnostic is too strong given the tested models are only ConvNexts and ViT

---

> ### Author Rebuttal · Authors · 2026-03-30
>
> **[Q1 & Q5] Importance of Unmergeability & Practicality of Trap$^2$**
>
> Thank you for raising this point. We agree that the practical scope can be stated more explicitly. In practice, a provider may wish to permit broad downstream recomposition for some releases, while keeping others standalone when they are tied to restricted, proprietary, or safety-sensitive capabilities. Trap² is intended for this provider-side, pre-release setting: the protection is embedded during fine-tuning and is aimed at discouraging indiscriminate downstream merging after release, rather than selectively approving or denying specific third-party merges. More selective or task-conditional protection would be an interesting extension, but is beyond the current scope. We will revise the relevant parts of the paper, especially the Introduction and Related Works sections, to make this practical scope and limitation more explicit.
>
> **[Q2 & Q6] Architecture Dependence of Post-hoc Protection Methods & Scope of the Architecture-Agnostic Claim**
>
> Thank you for raising this point. We agree that the scope of this claim can be stated more precisely. Our intention is not to claim that prior post-hoc protection methods are impossible to extend beyond Transformers. As discussed in the Introduction and Figure 1, and further detailed in Sections 2.2 and 4, the representative post-hoc methods considered here rely on Transformer-specific symmetries and full-weight access, and therefore are not directly applicable in the adapter-only and non-attention settings studied here.
>
> For Trap$^2$, the term “architecture-agnostic” refers to the level of the objective and formulation: the objective is defined at the level of the loss under scaled updates, rather than through architecture-specific symmetries. In that sense, the formulation is, in principle, not tied to a particular backbone family. This is also reflected in Section 5 (“Extension to Full Fine-Tuning”), which describes a unified formulation for adapter-based and full-checkpoint releases. Empirically, Section 6.2 validates the same qualitative behavior on ViT-B/32, ViT-L/14, and ConvNeXt. At the same time, we agree that this should not be read as a universal empirical claim over all architectures, and we will revise the paper to make this scope more explicit.
>
> **[Q3] More than $s = 1$ for the Nominal Scale**
>
> We appreciate this question. While $s = 1$ may appear overly special, $s$ in our formulation simply denotes the effective scaling coefficient of the released update, i.e., $W = W_0 + s \cdot \Delta W$; in LoRA, this corresponds to the effective scale $s = \alpha / r$ with $\Delta W = BA$. In our experiments, we use the standard standalone LoRA setting with $\alpha = r$, so $s = 1$ is the natural nominal reference point. More generally, the nominal scale depends on how the provider intends the released update to be used: for example, if a release is intended to operate at an effective scale $s = 2$, then $2$ would be the appropriate nominal point, and off-nominal behavior should be interpreted relative to that scale. We will revise the paper to clarify that $s = 1$ is not an arbitrary choice, but the standard standalone operating point in our setup and a commonly used LoRA convention, rather than a uniquely prescribed optimum.
>
> **[Q4] Beyond $s > 1$**
>
> Thank you for pointing this out. Figure 2 was intended to illustrate the key behavior of the objective over the most relevant normalized range. In the paper, the main merge proxy we emphasize is down-scaling (e.g., uniform averaging gives $s = 1/N$), which is the reason we visualized the $s < 1$ regime. However, we agree that this could have been stated more clearly, especially since the figure itself does not show how the behavior extends beyond the plotted range.
>
> To address this point directly, we additionally report the same experiment with $s$ extended up to $3.0$. As illustrated in Table A, while the unprotected model degrades gradually as $s$ increases, Trap$^2$ drops sharply beyond the nominal scale and remains near zero over a broad off-nominal range, making the trend for $s > 1$ explicit.
>
> |Method|1.0|1.1|1.2|1.3|1.4|1.5|1.6|1.7|1.8|1.9|2.0|2.1|2.2|2.3|2.4|2.5|2.6|2.7|2.8|2.9|3.0|
> |---|---:|---:|---:|---:|---:|---:|---:|---:|---:|---:|---:|---:|---:|---:|---:|---:|---:|---:|---:|---:|---:|
> |Unprotected|98.407|98.288|98.199|97.902|97.338|96.437|95.101|93.023|89.440|84.046|76.376|65.350|52.811|41.033|30.463|21.081|13.866|9.204|6.344|4.503|3.335|
> |Trap$^2$|98.911|34.778|0.020|0.000|0.000|0.000|0.010|0.020|0.020|0.030|0.030|0.049|0.069|0.109|0.188|0.238|0.277|0.435|0.881|1.633|2.524|
>
> [Table A] Extended scale sweep of GTSRB adapters beyond the plotted range in Figure 2.

---

> > ### Author Rebuttal · Reviewer_fUDN · 2026-04-04
> >
> > Thanks for the response. I will adjust the score accordingly.

---

> > > ### Author Response · Authors · 2026-04-07
> > >
> > > We are grateful for the time you put into both the original review and the rebuttal discussion. Your comments pushed us to sharpen several parts of the paper, especially the practical framing, the scope of the architecture-agnostic claim, and the interpretation of the scale analysis. We believe the paper is more precise because of that, and we will incorporate these improvements in the final version. If you have not yet had a chance to reflect the update in the system, we would greatly appreciate if you could do so when convenient.
> > >
> > > Sincerely,
> > >
> > > The Authors

---

### Decision · Program_Chairs · 2026-04-30

**Decision:**

Accept (regular)

**Comment:**

Scores settle at 3/6/5/2, with three reviewers positive and eaTZ maintaining rejection.
The paper addresses a governance gap with a sound, architecture-agnostic framework that works across non-Transformer architectures and both adapter and full-checkpoint releases. The rebuttal substantially expanded the empirical scope, with fUDN and aCEt declaring concerns fully resolved.

eaTZ's remaining concerns are hyperparameter sensitivity and limited LLM validation. These are real limitations worth stating clearly in the paper. The authors argue that λ tuning is a one-time pre-release cost rather than a deployment burden, and the LLM experiments demonstrate the same qualitative protection behavior extends beyond vision. The recovery concern, eaTZ's most substantive critique, is resolved favorably when the zero-shot baseline is included: the protected merge after SFT recovery reaches only 3 points above zero-shot versus 14 points for the unprotected baseline, consistent with the claimed protection. As Pb7J notes, formal robustness guarantees are unattainable in this domain, making empirical evidence the appropriate standard. The remaining gaps do not outweigh the contribution.